# Bayesian Causal Structural Learning with Zero-Inflated Poisson Bayesian Networks

**Junsouk Choi**
Department of Statistics
Texas A&M University
College Station, TX 77843
jchoi@stat.tamu.edu

**Robert Chapkin**
Department of Nutrition
Texas A&M University
College Station, TX 77843
r-chapkin@tamu.edu

**Yang Ni**
Department of Statistics
Texas A&M University
College Station, TX 77843
yni@stat.tamu.edu

## Abstract

Multivariate zero-inflated count data arise in a wide range of areas such as economics, social sciences, and biology. To infer causal relationships in zero-inflated count data, we propose a new zero-inflated Poisson Bayesian network (ZIPBN) model. We show that the proposed ZIPBN is identifiable with cross-sectional data. The proof is based on the well-known characterization of Markov equivalence class which is applicable to other distribution families. For causal structural learning, we introduce a fully Bayesian inference approach which exploits the parallel tempering Markov chain Monte Carlo algorithm to efficiently explore the multi-modal network space. We demonstrate the utility of the proposed ZIPBN in causal discoveries for zero-inflated count data by simulation studies with comparison to alternative Bayesian network methods. Additionally, real single-cell RNA-sequencing data with known causal relationships will be used to assess the capability of ZIPBN for discovering causal relationships in real-world problems.

## 1 Introduction

This paper addresses causal structural learning problems for zero-inflated count data. While true causality can only be determined using controlled experimentation (Gourévitch *et al.*, 2006), statistical methods are useful in generating testable causal hypotheses which can semi-automate and expedite the causal discovery process. Bayesian networks (BNs) are popular approaches for causal structural learning and inference (Pearl, 2009). However, BNs may not be identifiable with cross-sectional data due to Markov equivalence class (MEC, Heckerman *et al.* 1995) in which all BNs encode the same conditional independence assertions. This implies that without further assumptions, one cannot differentiate BNs that belong to the same MEC.

Several approaches have been developed to address the identifiability issue of BNs. BNs for continuous data are often represented as sparse additive noise models. Under such representation, BNs are identifiable if the noises are non-Gaussian (Shimizu *et al.*, 2006), if the functional form of the additive noise model is nonlinear plus very mild additional conditions (Hoyer *et al.*, 2009; Zhang & Hyvärinen, 2009), or if the noise variances are equal (Peters & Bühlmann, 2014). When the noises are non-Gaussian (Shimizu *et al.*, 2006), a BN is identifiable because it is equivalent to the independent component analysis (ICA) and hence its identifiability theory applies. To learn the causal structure, they extended the ICA algorithm with additional steps of rotation, normalization, and test-based pruning. Hoyer *et al.* (2009) proposed to learn the structure of a nonlinear BN by testing whether the noises are independent and showed that nonlinear BNs are identifiable for bivariate case. Zhang & Hyvärinen (2009) further considered additional nonlinear effects of measurement errors and proposed an ICA-based algorithm for bivariate BNs and a two-step approach (first identifying MEC by conditional independence tests and then determining a unique BN with a similar approach in Hoyer *et al.*

2009) for multivariate BNs. Peters *et al.* (2014) formally established the identifiability conditions for general multivariate additive noise models and developed test-based and score-and-greedy-search learning algorithms.

Identifiability of BNs for count data is less studied. Recent work by Park & Raskutti (2015) proposed a Poisson BN and showed that it is identifiable based on the overdispersion properties of Poisson BNs. Their learning algorithm consists of two steps: first obtaining an ordering of the variables with overdispersion scoring and then estimating the causal structure given the ordering. By replacing overdispersion score with moment ratio score, Park & Park (2019) extended Poisson BNs to the generalized hypergeometric family which contains many count distributions such as binomial, Poisson, and negative binomial.

This paper is motivated by causal structural learning for zero-inflated count data which arise in a wide range of areas such as educational psychology (Fox, 2013), genomic experiments (Kang *et al.*, 2011), ecology (Barry & Welsh, 2002), behavior studies (Hua *et al.*, 2014), and economics (Staub & Winkelmann, 2013). Particularly, we consider an application in reverse-engineering causal gene regulatory networks from single-cell RNA-sequencing (scRNA-seq) data. scRNA-seq data are zero-inflated due to the detection limit of the current techniques in capturing the low amounts of mRNA in individual cells. Standard distributions such as Gaussian and Poisson cannot adequately capture the excessive zeros in zero-inflated count data.

To explicitly account for zero-inflation, we propose a zero-inflated Poisson Bayesian network (ZIPBN) model. While generalized hypergeometric BNs are a fairly general class of discrete BNs, the proposed ZIPBN is not a special case. We establish the identifiability theory for the proposed ZIPBN without assuming causal faithfulness. The proof is based on the characterization of MEC, and is applicable to other distribution families. We introduce a fully Bayesian learning approach based on Markov chain Monte Carlo (MCMC) to explore the network space. Compared to the vast majority of the existing casual BN learning algorithms, the proposed Bayesian approach has the advantage of being able to naturally quantify the uncertainties associated with the estimated networks via posterior distributions. Moreover, unlike heuristic/greedy search algorithms, MCMC is theoretically guaranteed to converge to its stationary distribution (i.e., the targeted posterior distribution). The Bayesian formulation also automatically guards against the multiplicity problem often experienced by test-based algorithms. Practically, we adopt parallel tempering technique (Geyer, 1991) for efficient exploration of the multi-modal network space. We demonstrate the utility of the proposed ZIPBN in causal discoveries using synthetic data and real scRNA-seq data with known causal relationships.

## 2 Zero-inflated Poisson Bayesian networks

### 2.1 Sampling model

Let $\boldsymbol{X} = \{X_1, \ldots, X_p\}$ denote a set of $p$ random zero-inflated counts, of which the causal relationships will be investigated with BNs. A BN $\mathcal{B} = (\mathcal{G}, \boldsymbol{\theta})$ consists of two parts: a directed acyclic graph (DAG) $\mathcal{G}$ and a set of parameters $\boldsymbol{\theta}$ associated with the DAG. A DAG $\mathcal{G} = (V, \boldsymbol{E})$ is defined by a set of nodes $V = \{1, \ldots, p\}$, representing the random variables in $\boldsymbol{X}$, and a set of arrows or directed edges $\boldsymbol{E} = [e_{jk}]_{j,k}$ such that $e_{jk} = 1$ if $k \rightarrow j$, representing causal relationships between the nodes. Matrix $\boldsymbol{E}$ is also known as the adjacency matrix of graph $\mathcal{G}$. Node $k$ is said to be a parent of $j$ if $k \rightarrow j$. Denote the set of all the parents of $j$ by $pa(j) = \{k \in V : k \rightarrow j\}$. For example, in Figure 1(d), $pa(2) = \{1, 3\}$ and $pa(1) = \emptyset$. DAGs allow no directed cycle (i.e., one cannot return to the same node by following the arrows) and the acyclic assumption leads to a convenient factorization of the joint distribution of $\boldsymbol{X}$ into a product of conditional distributions,

$$p\left(\boldsymbol{X}\right) = \prod_{j=1}^{p} p\left(X_j \,\big|\, \boldsymbol{X}_{pa(j)}\right), \tag{1}$$

where $\boldsymbol{X}_{pa(j)} = \{X_k : k \in pa(j)\}$. To explicitly account for excessive zero counts, we assume each conditional distribution in the factorization (1) to be a zero-inflated Poisson regression,

$$Pr\left(X_j = x \,\big|\, \boldsymbol{X}_{pa(j)}\right) = \begin{cases} \eta_j + (1 - \eta_j)\exp\left(-\lambda_j\right) & \text{if} \quad x = 0 \\ (1 - \eta_j)\frac{\exp(-\lambda_j)\lambda_j^x}{x!} & \text{if} \quad x > 0, \end{cases} \tag{2}$$

where $\log\{\eta_j/(1-\eta_j)\} = \sum_{k \in pa(j)} \alpha_{jk} X_k + \delta_j$ and $\log(\lambda_j) = \sum_{k \in pa(j)} \beta_{jk} X_k + \gamma_j$. The parameter $\eta_j$ accounts for the extra zeros in addition to the zeros that arise from the Poisson component, while the parameter $\lambda_j$ is the rate parameter of the Poisson component. It is clear from (2) that $k \notin pa(j)$ if and only if $\alpha_{jk} = \beta_{jk} = 0$. Therefore, learning the graph structure is equivalent to finding which pairs $(\alpha_{jk}, \beta_{jk})$ are (not) zeros. Let $\boldsymbol{\alpha} = \{\alpha_{jk}\}_{j,k}, \boldsymbol{\beta} = \{\beta_{jk}\}_{j,k}, \boldsymbol{\delta} = \{\delta_j\}_j, \boldsymbol{\gamma} = \{\gamma_j\}_j$, and let $\boldsymbol{\theta} = \{\boldsymbol{\alpha}, \boldsymbol{\beta}, \boldsymbol{\delta}, \boldsymbol{\gamma}\}$. Models (1) and (2) define the sampling distribution of the proposed ZIPBN parameterized by the graph structure $\boldsymbol{E}$ and the associated parameters $\boldsymbol{\theta}$.

## 2.2 Identifiability

BNs in the same MEC are generally not distinguishable from each other as they have exactly the same Markov properties (i.e., conditional independence assertions). For instance, despite the seemingly different graph structures, the BNs in Figure 1(a)-(c) encode the same conditional independence of $X_1$ and $X_3$ given $X_2$ whereas the BN in (d) encodes the marginal independence of $X_1$ and $X_3$. No additional independence can be read off from any of these BNs. What it implies in graph structural learning is that BNs are only identifiable up to MEC without further distributional assumptions. For example, the well-known nonparametric PC algorithm (Spirtes *et al.*, 2000) only outputs the best MEC rather than individual BNs. However, even with additional distributional assumptions, BNs may still be non-identifiable due to distribution equivalence (Spirtes & Zhang, 2016): two BNs $\mathcal{B}_1 = (\mathcal{G}_1, \boldsymbol{\theta}_1)$ and $\mathcal{B}_2 = (\mathcal{G}_2, \boldsymbol{\theta}_2)$ are distribution equivalent if for any parameter values $\boldsymbol{\theta}_1$ of $\mathcal{B}_1$ there exists parameter values $\boldsymbol{\theta}_2$ of $\mathcal{B}_2$ that represents the same distribution, and vice versa. Gaussian and binary BNs are two examples of distribution equivalent BNs and hence are non-identifiable.

The identifiability issue of discrete BNs has been addressed in prior work (Park & Raskutti, 2015; Park & Park, 2019) for Poisson and generalized hypergeometric family under the *causal sufficiency* assumption that all relevant variables have been observed. However, the identifiability property of the ZIPBN has not been studied and we will show that the proposed ZIPBN is identifiable with a different proof technique from those in prior work, which can also be used to prove identifiability for other distribution families. Our proof relies on a well-known characterization of MEC, i.e., Markov equivalent BNs must have the same skeleton – the skeleton of a BN is the undirected graph resulting from converting all directed edges to undirected edges.

**Theorem 1.** *Assume causal sufficiency, ZIPBNs are identifiable (i.e., no two ZIPBNs are distribution equivalent).*

The main idea behind the proof is to show that any two Markov equivalent ZIPBNs $\mathcal{B}_1 = (\mathcal{G}_1, \boldsymbol{\theta}_1)$ and $\mathcal{B}_2 = (\mathcal{G}_2, \boldsymbol{\theta}_2)$ are not distribution equivalent. Without loss of generality, we assume variables are sorted according to the perfect ordering of $\mathcal{G}_1$ so that if $k \to j$ then $k$ must appear before $j$. For example, the BN in Figure 1(a) is complied with the perfect ordering whereas those in Figures 1(b)-(d) are not. We first show that the parent set $pa(p)$ of the last node $p$ has to be identical in $\mathcal{G}_1$ and $\mathcal{G}_2$ if they have the same skeleton and they are distribution equivalent. Then we use mathematical induction to show that this is true for any node and therefore $\mathcal{G}_1$ and $\mathcal{G}_2$ have to be identical. The detailed proof is provided in the Supplementary Material. As an illustration of the generality of our proof, we adapt it for Poisson BN (different from that in Park & Raskutti 2015) which is shown in the Supplementary Material as well.

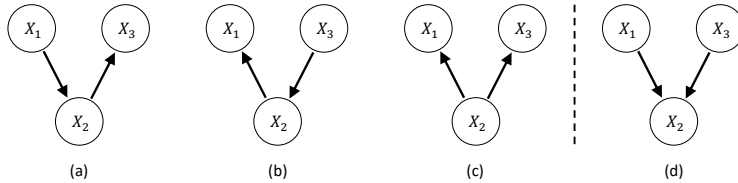

(a)　　　　　(b)　　　　　(c)　　　　　(d)

Figure 1: Examples of Bayesian networks with three nodes. The BNs in (a)-(c) are Markov equivalent and form a Markov equivalence class ($X_1 \perp X_3 | X_2$). The BN in (d) forms another Markov equivalence class ($X_1 \perp X_3$).

Note that we do not assume *causal faithfulness* which is adopted by many existing BN learning algorithms (Chickering, 2002; Peters & Bühlmann, 2014). A distribution $p(\cdot)$ is *faithful* to the causal graph $\mathcal{G}$ if $\mathcal{G}$ encodes all the conditional independencies in $p(\cdot)$. Faithfulness can be violated with a limited sample size (Uhler *et al.*, 2013) or in an equilibrium-maintaining system such as a biological

system. While both Gaussian and multinomial BNs can have accidental cancellation of positive and negative effects and therefore become unfaithful, the proposed ZIPBN does not allow such cancellation because of its count nature; also see Park & Park (2019).

## 2.3 Prior model

In this section, we introduce prior distributions of $\boldsymbol{E}$ and $\boldsymbol{\theta}$ which allow for a fully Bayesian learning approach to infer sparse ZIPBN.

Prior of $\boldsymbol{E}$. We assume a Bernoulli prior for each edge $e_{jk}$ with edge-inclusion probability $\rho$ subject to the constraint that the resulting graph $\mathcal{G}$ is DAG,

$$p(\boldsymbol{E}|\rho) = z(\rho)^{-1} \prod_{j \neq k} \rho^{e_{jk}} (1-\rho)^{1-e_{jk}} I(\mathcal{G} \in \mathcal{D}), \tag{3}$$

where $I(\cdot)$ is a binary indicator function (=1 if the input is true and 0 otherwise), $\mathcal{D}$ is the collection of all DAGs with $p$ nodes, and $z(\rho)$ is the normalizing constant.

A hyperprior proportional to the product of a beta distribution and the normalizing constant $z(\rho)$ is defined for the edge-inclusion probability $\rho$:

$$p(\rho) \propto z(\rho)\rho^{a_\rho-1}(1-\rho)^{b_\rho-1}. \tag{4}$$

Including $z(\rho)$ in the prior of $p(\rho)$ serves to cancel out $z(\rho)^{-1}$ in (3) so that the full conditional of $\rho$ is beta distribution. Similar cancellation trick has been used and thoroughly investigated in Bayesian graphical lasso (Wang *et al.*, 2012).

Bayesian structural learning has a built-in penalty (also known as Bayesian Ockham's razor) for network complexity naturally induced by marginal likelihood with $\boldsymbol{\theta}$ integrated out (Jefferys & Berger, 1992). Moreover, the beta-like prior of $\rho$ allows for automatic multiplicity control in a way similar to beta prior for linear regression model (Scott & Berger, 2010). The additional multiplicity penalty is contained in the marginal prior of $\boldsymbol{E}$ (the analytical integration is provided in the Supplementary Material),

$$p(\boldsymbol{E}) \propto B\left(\sum_{j \neq k} e_{jk} + a_\rho, \sum_{j \neq k}(1-e_{jk}) + b_\rho\right) I(\mathcal{G} \in \mathcal{D}), \tag{5}$$

where $B(\cdot, \cdot)$ is the beta function. Unlike the penalty induced by the marginal likelihood, the multiplicity penalty increases with the number of variables $p$ and hence guards against false discoveries as graph size increases. For example, the marginal prior (5) favors the empty graph over a graph with one edge by a factor of $p^2 - p$ when $a_\rho = b_\rho = 1$ which clearly increases in $p$.

Prior of $\boldsymbol{\theta}$. Conditional on the graph structure $\boldsymbol{E}$, we assume a spike-and-slab prior on the edge strength $(\alpha_{jk}, \beta_{jk})$. Spike-and-slab prior is a two-component mixture distribution. The first component is a bivariate normal distribution indicating the presence of a significant causal relationship $e_{jk} = 1$, while the second component is a point mass at $\boldsymbol{0} = (0,0)$ representing a missing edge $e_{jk} = 0$. Let $\mathrm{N}_2(\boldsymbol{0}, \boldsymbol{P}^{-1})$ denote a centered bivariate normal density with precision matrix $\boldsymbol{P}$ and let $\delta_{\boldsymbol{0}}$ denote the Dirac measure at $\boldsymbol{0}$. Then the spike-and-slab prior is given by,

$$(\alpha_{jk}, \beta_{jk})|e_{jk}, \tau_1, \tau_2 \sim e_{jk}\mathrm{N}_2(\boldsymbol{0}, \boldsymbol{P}^{-1}) + (1 - e_{jk})\delta_{\boldsymbol{0}} \quad \text{for} \quad j \neq k,$$

where $\boldsymbol{P} = \mathrm{diag}(\tau_1, \tau_2)$ is a diagonal matrix.

For the intercepts $\delta_j$ and $\gamma_j$, we assign normal priors $\delta_j|\tau_3 \sim \mathrm{N}(0, \tau_3^{-1})$ and $\gamma_j|\tau_4 \sim \mathrm{N}(0, \tau_4^{-1})$, where $\tau_3$ and $\tau_4$ are the precision parameters. That is, the intercepts are not subject to selection. Finally, we assume that $\tau_\ell$ follows a gamma distribution, $\tau_\ell \sim \mathrm{gamma}(a_\tau, b_\tau)$ for $\ell = 1, 2, 3, 4$, to complete the hierarchical formulation of our model.

## 2.4 Posterior inference

The sampling distribution (Section 2.1) and the prior distribution (Section 2.3) will be combined as posterior distribution,

$$\pi(\boldsymbol{E}, \boldsymbol{\theta}, \boldsymbol{\psi}|\boldsymbol{X}) \propto p(\boldsymbol{X}|\boldsymbol{E}, \boldsymbol{\theta})p(\boldsymbol{\theta}|\boldsymbol{E}, \boldsymbol{\tau})p(\boldsymbol{E}|\rho)p(\boldsymbol{\tau})p(\rho), \tag{6}$$

which reflects one's updated belief regarding the unknown graph structure $\boldsymbol{E}$, the associated parameters $\boldsymbol{\theta}$, and hyperparameters $\boldsymbol{\psi} = \{\rho, \boldsymbol{\tau}\}$ with $\boldsymbol{\tau} = \{\tau_1, \tau_2, \tau_3, \tau_4\}$. The posterior distribution enables us to not only obtain a point estimate of the structure $\boldsymbol{E}$ but also naturally quantify the associated uncertainties through e.g., the posterior probability of edge inclusion $p(e_{jk} = 1|\boldsymbol{X})$.

Since the posterior distribution in (6) is analytically intractable, we use an MCMC sampler to draw posterior samples. The space of the causal structure $\boldsymbol{E}$ is discrete and high dimensional, resulting in a multi-modal posterior distribution. MCMC samplers relying on local moves such as Gibbs samplers are easily trapped in local modes, which yields inefficient exploration of the network space. To alleviate the multi-modal issue, we design a parallel-tempered MCMC (Geyer, 1991).

Specifically, we run $M$ Markov chains in parallel with different, but related, target distributions $\pi_1, \ldots, \pi_M$. These target distributions are defined as fractional power of the posterior distribution $\pi_m \propto \pi(\boldsymbol{E}, \boldsymbol{\theta}, \boldsymbol{\psi}|\boldsymbol{X})^{1/T_m}$ for $m = 1, \ldots, M$ where $1 = T_1 < T_2 < \cdots < T_M$ are a sequence of increasing "temperatures". Clearly, the first target distribution $\pi_1 = \pi(\boldsymbol{E}, \boldsymbol{\theta}, \boldsymbol{\psi}|\boldsymbol{X})$ with the lowest temperature $T_1 = 1$ (the cold chain) is our desired posterior distribution. The parallel tempering technique utilizes the other $M - 1$ heated chains to avoid being trapped in local modes. This is because the fractional power (for temperature $T_m > 1$) flattens the distribution $\pi_m$, which makes it easier to move between local modes and explore the multi-modal network space. At each iteration of the MCMC, we perform one of the two updating steps, a swapping step with probability $p_s$ or a Gibbs step with probability $1 - p_s$.

**Swapping**. We randomly pick two chains $\ell$ and $m$ and propose to swap their states. The swapping is accepted with probability $\min\{1, R_s\}$ where $R_s$ is the Metropolis-Hasting (MH) acceptance ratio,

$$R_s = \frac{\pi(\boldsymbol{E}_\ell, \boldsymbol{\theta}_\ell, \boldsymbol{\psi}_\ell|\boldsymbol{X})^{1/T_m} \pi(\boldsymbol{E}_m, \boldsymbol{\theta}_m, \boldsymbol{\psi}_m|\boldsymbol{X})^{1/T_\ell}}{\pi(\boldsymbol{E}_\ell, \boldsymbol{\theta}_\ell, \boldsymbol{\psi}_\ell|\boldsymbol{X})^{1/T_\ell} \pi(\boldsymbol{E}_m, \boldsymbol{\theta}_m, \boldsymbol{\psi}_m|\boldsymbol{X})^{1/T_m}},$$

where the subscript of each parameter indicates which chain it comes from. The swapping helps chains with lower temperatures mix better, but induces dependence across the chains which makes each chain no longer Markov. However, the $M$ chains together form a Markov chain with the target distribution $\Pi = \pi_1 \times \cdots \times \pi_M$. The Monte Carlo samples from the cold chain target the marginal distribution $\pi_1$ of $\Pi$ and therefore the right posterior distribution.

**Gibbs**. We update all the parameters $\boldsymbol{E}, \boldsymbol{\theta} = \{\boldsymbol{\alpha}, \boldsymbol{\beta}, \boldsymbol{\delta}, \boldsymbol{\gamma}\}$, and $\boldsymbol{\psi} = \{\rho, \boldsymbol{\tau}\}$ in each chain independently. The Metropolis-within-Gibbs approach is employed to update the causal structure $\boldsymbol{E}$ and model parameters $\boldsymbol{\theta}$, while the hyperparameters $\boldsymbol{\psi}$ are updated based on their full conditional distributions. We proceed with the sequence: (i) $[\boldsymbol{E}, \boldsymbol{\theta} \mid \boldsymbol{\psi}]$, (ii) $[\boldsymbol{\theta} \mid \boldsymbol{E}, \boldsymbol{\psi}]$, and (iii) $[\boldsymbol{\psi} \mid \boldsymbol{E}, \boldsymbol{\theta}]$.

(i) $[\boldsymbol{E}, \boldsymbol{\theta} \mid \boldsymbol{\psi}]$. When updating $\boldsymbol{E}$, we need to jointly update $\boldsymbol{\theta}$ at the same time because a change in $\boldsymbol{E}$ alters the dimension of $\boldsymbol{\theta}$. Let $\boldsymbol{E}^\star$ and $\boldsymbol{\theta}^\star = \{\boldsymbol{\alpha}^\star, \boldsymbol{\beta}^\star, \boldsymbol{\delta}^\star, \boldsymbol{\gamma}^\star\}$ denote new parameter values that we propose. They are accepted with probability $\min\{1, R_{\boldsymbol{E}}^m\}$ with

$$R_{\boldsymbol{E}}^m = \frac{\{p(\boldsymbol{X}|\boldsymbol{E}^\star, \boldsymbol{\theta}^\star)p(\boldsymbol{\theta}^\star|\boldsymbol{E}^\star, \boldsymbol{\tau})p(\boldsymbol{E}^\star|\rho)\}^{1/T_m} q(\boldsymbol{E}, \boldsymbol{\theta}|\boldsymbol{E}^\star, \boldsymbol{\theta}^\star)}{\{p(\boldsymbol{X}|\boldsymbol{E}, \boldsymbol{\theta})p(\boldsymbol{\theta}|\boldsymbol{E}, \boldsymbol{\tau})p(\boldsymbol{E}|\rho)\}^{1/T_m} q(\boldsymbol{E}^\star, \boldsymbol{\theta}^\star|\boldsymbol{E}, \boldsymbol{\theta})},$$

where $q(\cdot \mid \cdot)$ denotes a proposal density. The specific form of the proposal is obtained by randomly selecting one of the following two schemes with equal probabilities.

1. *Birth or death*: For $j \neq k$, update one edge $e_{jk}$ at a time jointly with $(\alpha_{jk}, \beta_{jk}, \delta_j, \gamma_j)$. We propose a new state $e_{jk}^\star$ of $e_{jk}$, either from 0 to 1 (birth of an edge) or from 1 to 0 (death of an edge). If the resulting $\boldsymbol{E}^\star$ contains a directed cycle, go to next step. If $e_{jk}^\star = 1$, we generate $\alpha_{jk}^\star$ and $\beta_{jk}^\star$ from independent normal distributions centered at zero; otherwise we set $\alpha_{jk}^\star = \beta_{jk}^\star = 0$. New states $(\delta_j^\star, \gamma_j^\star)$ are proposed using Gaussian random walks.

2. *Reversal*: Update $(e_{jk}, e_{kj}, \alpha_{jk}, \alpha_{kj}\beta_{jk}, \beta_{kj}, \delta_j, \delta_k, \gamma_j, \gamma_k)$ for each $(j, k) \in \{(j, k) : e_{jk} = 1\}$ sequentially. We form new states $(e_{jk}^\star, e_{kj}^\star)$ by flipping $e_{jk}$ from 1 to 0 and $e_{kj}$ from 0 to 1 (reversal of an arrow). If it creates a directed cycle, go to next step. Otherwise, we propose $\alpha_{kj}^\star$ and $\beta_{kj}^\star$ from independent centered normal distributions with and set $\alpha_{jk}^\star = \beta_{jk}^\star = 0$. Gaussian random walks are used to generate new states $(\delta_j^\star, \delta_k^\star, \gamma_j^\star, \gamma_k^\star)$.

(ii) $[\boldsymbol{\theta} \mid \boldsymbol{E}, \boldsymbol{\psi}]$. Conditional on $\boldsymbol{E}$, we update $\boldsymbol{\alpha}$ through Gaussian random walk. We propose a new state $\alpha_{jk}^\star$ from the normal distribution centered at $\alpha_{jk}$ and accept it with

$\min\left[1, \left\{\frac{p(\boldsymbol{X}|\boldsymbol{E},\boldsymbol{\alpha}^\star,\boldsymbol{\beta},\boldsymbol{\delta},\boldsymbol{\gamma})p(\boldsymbol{\alpha}^\star|\boldsymbol{E})}{p(\boldsymbol{X}|\boldsymbol{E},\boldsymbol{\alpha},\boldsymbol{\beta},\boldsymbol{\delta},\boldsymbol{\gamma})p(\boldsymbol{\alpha}|\boldsymbol{E})}\right\}^{1/T_m}\right]$ for each $(j,k)$ such that $e_{jk} = 1$. Parameters $\boldsymbol{\beta},\boldsymbol{\delta},\boldsymbol{\gamma}$ are updated in a similar way.

(iii) $[\boldsymbol{\psi} \mid \boldsymbol{E},\boldsymbol{\theta}]$. The precision parameters $\boldsymbol{\tau}$ are drawn from their full conditional distributions, $\tau_1 \sim$ gamma$(\sum_{j\neq k} e_{jk}/2 + a_\tau, \sum_{j\neq k} \alpha_{jk}^2/2 + b_\tau)$, $\tau_2 \sim$ gamma$(\sum_{j\neq k} e_{jk}/2 + a_\tau, \sum_{j\neq k} \beta_{jk}^2/2 + b_\tau)$, $\tau_3 \sim$ gamma$(p/2 + a_\tau, \sum_j \delta_j^2/2 + b_\tau)$, and $\tau_4 \sim$ gamma$(p/2 + a_\tau, \sum_j \gamma_j^2/2 + b_\tau)$. Likewise, $\rho$ is sampled from its full conditional distribution, beta$(\sum_{j\neq k} e_{jk} + a_\rho, \sum_{j\neq k}(1 - e_{jk}) + b_\rho)$.

For each Markov chain, the worst-case per iteration cost is $O(np^2)$ mainly due to the likelihood evaluation, which is reduced to $O(\max(n,p)p)$ for sparse networks (i.e., $|\boldsymbol{E}| = O(p)$). The complete MCMC algorithm is provided as pseudo-code in Algorithm 1. In the algorithm, *parfor* indicates a parallelizable loop, while *for* indicates a sequential loop. The code implementing the MCMC is

---

**Algorithm 1** Parallel-Tempered MCMC for ZIPBN

1: **Input:** data $\boldsymbol{X}$, hyperparameters $(a_\rho, b_\rho, a_\tau, b_\tau)$, temperatures $1 = T_1 < \cdots < T_M$, swapping probability $p_s$, and number of iterations $N$
2: Initialize all the parameters for every chain $\{\boldsymbol{E}_m^{(0)}, \boldsymbol{\theta}_m^{(0)}, \boldsymbol{\psi}_m^{(0)}\}_{m=1}^M$
3: **for** $i$ in $1,\ldots,N$ **do**
4:      Draw a Bernoulli random variable $u$ with probability $p_s$
5:      **if** $u = 1$ **then**
6:          Perform a swapping step to swap $\{\boldsymbol{E}_m^{(i)}, \boldsymbol{\theta}_m^{(i)}, \boldsymbol{\psi}_m^{(i)}\}$ and $\{\boldsymbol{E}_\ell^{(i)}, \boldsymbol{\theta}_\ell^{(i)}, \boldsymbol{\psi}_\ell^{(i)}\}$
7:      **else**
8:          **parfor** $m$ in $1,\ldots,M$ **do**
9:              Perform a Gibbs step for chain $m$ to update $\boldsymbol{E}_m^{(i)}, \boldsymbol{\theta}_m^{(i)}, \boldsymbol{\psi}_m^{(i)}$
10:          **end parfor**
11:      **end if**
12: **end for**
13: **Output:** Monte Carlo samples from the cold chain, $\{\boldsymbol{E}_1^{(i)}, \boldsymbol{\theta}_1^{(i)}, \boldsymbol{\psi}_1^{(i)}\}_{i=1}^N$

---

available in the Supplementary Material.

At the completion of MCMC, we retain the samples from the cold chain,

$$\boldsymbol{E}^{(1)}, \boldsymbol{\theta}^{(1)}, \boldsymbol{\psi}^{(1)}, \ldots, \boldsymbol{E}^{(N)}, \boldsymbol{\theta}^{(N)}, \boldsymbol{\psi}^{(N)},$$

where the superscript is the sample index. To infer the causal structure $\boldsymbol{E}$, we compute the marginal posterior inclusion probabilities $p_{jk} = p(e_{jk} \mid \boldsymbol{X}) \approx N^{-1}\sum_{i=1}^N e_{jk}$. For a point estimate $\widehat{\boldsymbol{E}} = [\widehat{e}_{jk}]$, we set $\widehat{e}_{jk} = I(p_{jk} > c)$ where $c$ is a pre-specified cutoff. We set the cutoff at $c = 0.5$ in the simulation for simplicity. In the application, the cutoff is chosen to control the posterior expected false discovery rate (Müller *et al.*, 2006). Posterior estimates of $\boldsymbol{\theta}$ and $\boldsymbol{\psi}$ can be easily obtained conditional on $\widehat{\boldsymbol{E}}$.

We remark that MCMC-based BN learning algorithm has at least two prominent advantages over heuristic/greedy search algorithms (e.g., Chickering (2002)). First, the posterior inference based on the proposed MCMC allow us to naturally quantify the uncertainties associated with the estimated causal networks. The uncertainty quantification is especially important in BN learning, as multiple BNs may explain the data equally well with a limited sample size and hence point estimates are often not satisfactory. Second, the convergence of MCMC algorithms requires a sufficient number of MCMC iterations whereas the convergence of heuristic/greedy search algorithms often requires a sufficiently large sample size. In practice, it is often infeasible and expensive to obtain a large enough sample size relatively to the super-exponential size of DAG space. However, it is comparatively much easier and cheaper to increase the size of MCMC. Additionally, the practical convergence of MCMC can be monitored via various diagnostics, e.g., Gelman-Rubin's potential scale reduction factor (Gelman *et al.*, 1992).

# 3 Experiments

## 3.1 Simulations

We empirically evaluated the causal discovery of the proposed ZIPBN with synthetic data. We simulated data under different samples sizes $n \in \{250, 500, 1000\}$ and different numbers of nodes $p \in \{25, 50, 75\}$. A sparse DAG was randomly generated with $p$ edges and the causal structure was assumed to be constant for each simulation setting. Given the DAG, we generated non-zero elements of the edge-specific parameters $(\alpha_{jk}, \beta_{jk})$ from independent uniform distributions: $\alpha_{jk} \sim \text{U}(0.3, 1)$ and $\beta_{jk} \sim \text{U}(-1, -0.3)$. Likewise, the intercepts $\delta_j$ and $\gamma_j$ were also generated uniformly $\delta_j \in \text{U}(-2, -1)$ and $\gamma_j \in \text{U}(1, 2)$. These ranges were chosen so that the resulting observations were not all zeros or extremely large values. The resulting observations have $\sim 50\%$ zeros. Each simulation setting was repeated 30 times.

For comparison, we considered two alternative discrete BN learning algorithms: the OverDispersion Scoring (ODS) algorithm for Poisson BNs (Park & Raskutti, 2015) and the Moments Ratio Scoring (MRS) algorithm for the generalized hypergeometric BNs (Park & Park, 2019).

For the proposed ZIPBN, we used non-informative prior by setting the hyperparameters to be $(a_\tau, b_\tau) = (0.01, 0.01)$ and $(a_\rho, b_\rho) = (0.5, 0.5)$ which are commonly used in Bayesian variable selection and have negligible influence on the posterior inference. We ran $M = 10$ parallel chains for $3,000$ iterations, of which the first $1,500$ iterations were discarded as burn-in. The temperatures were chosen uniformly between 0 and 1 on the log-scale, i.e., $\log(T_m) = (m-1)/9$ for $m = 1, \ldots, 10$. The swapping probability $p_s$ was chosen to be 10%. To summarize the operating characteristics, we calculated the true positive rate (TPR), the false discovery rate, and the Mattews correlation coefficient (MCC) defined as

$$\text{MCC} = \frac{\text{TP} \times \text{TN} - \text{FP} \times \text{FN}}{\sqrt{(\text{TP} + \text{FP})(\text{TP} + \text{FN})(\text{TN} + \text{FP})(\text{TN} + \text{FN})}},$$

where TP, TN, FP, and FN denote the true positives, true negatives, false positives, and false negatives, respectively. MCC takes value between -1 and 1 with 1 being perfect selection and 0 being random guess.

The operating characteristics of each scenario are summarized in Table 1. The proposed ZIPBN consistently outperformed both ODS and MRS across scenarios, indicating the inadequacy of Poisson and hyper-Poisson in modeling zero-inflated count data. Although MRS had similar, sometimes even slightly higher, TPR compared to ZIPBN, FDR of MRS was always substantially higher than that of ZIPBN, resulting in lower MCC. Additionally, as the network size increased, FDR of MRS increased rapidly, while FDR of the proposed ZIPBN was much better controlled. This result demonstrated the power of automatic multiplicity adjustment of our priors in (3)-(5). As expected, ODS did not work well due to lack of flexibility in accommodating the zero-inflation. Note that as the sample size $n$ increases from 500 to $1,000$, the average operating characteristics of ZIPBN became slightly worse. This is because larger sample size leads to peakier local modes which can be alleviated by increasing the number of parallel chains or the number of MCMC iterations (results not shown).

Next, we performed additional simulations to assess ZIPBN under different percentages of zeros. Three different sets of true parameters are produced to generate datasets having $\sim 25\%$, $\sim 50\%$, and $\sim 75\%$ zeros for $n = 500$ and $p = 50$. The simulation results are reported in Table 2. ZIPBN clearly outperformed both ODS and MRS in every case. As the percentage of zeros increased up to $\sim 75\%$, the overall performance of ZIPBN did not deteriorate much while FDR of MRS was doubled.

Lastly, we tested robustness of ZIPBN on misspecified models. We considered (i) Poisson BN that does not include a zero-inflation component and (ii) zero-inflated negative binomial (ZI-NegBin) BN in which the Poisson distribution of ZIPBN is replaced by a more flexible negative binomial distribution. Since Poisson BN is a special case of ZIPBN, we also tested the Bayesian inference algorithm for Poisson BN by fixing $\eta_j = 0$ in the proposed MCMC, denoted by BayPBN hereafter. Again, ODS and MRS were used as benchmarks. For Poisson BN, we generated the data with $n = 500$ and $p = 50$ in the same way as before except that the zero-inflation probability $\eta_j$ was fixed at 0. For ZI-NegBin BN, we generated the data with $\sim 50\%$ zeros and dispersion parameters uniformly drawn from $(1, 5)$. The results are shown in Table 3. When Poisson BN is the true model, BayPBN was clearly the best and ZIPBN still outperformed (higher TPR and lower FDR) ODS and MRS even though the data are not zero-inflated, which demonstrates the utility of Bayesian inference

for Bayesian networks. In the scenario of ZI-NegBin BN, the overall performance of ZIPBN was still better than MRS, but with a smaller gap (MCC 0.540 vs 0.455). We anticipate MRS to outperform ZIPBN as the percentage of zeros goes to 0.

Table 1: Average operating characteristics over 30 simulations for each zero-inflated scenario. The standard error for each statistic is given in parentheses. The best performance is in boldface.

| Method | | $p = 50$ | | | $n = 1000$ | | |
| | | $n$ | | | $p$ | | |
| | | 250 | 500 | 1000 | 25 | 50 | 75 |
|---|---|---|---|---|---|---|---|
| ZIPBN | | | | | | | |
| | TPR | **0.813 (0.010)** | **0.839 (0.010)** | 0.811 (0.007) | 0.851 (0.014) | 0.811 (0.007) | **0.750 (0.012)** |
| | FDR | **0.178 (0.011)** | **0.180 (0.010)** | **0.246 (0.009)** | **0.186 (0.016)** | **0.246 (0.009)** | **0.267 (0.013)** |
| | MCC | **0.814 (0.011)** | **0.826 (0.010)** | **0.777 (0.008)** | **0.825 (0.015)** | **0.777 (0.008)** | **0.738 (0.013)** |
| ODS | | | | | | | |
| | TPR | 0.403 (0.006) | 0.452 (0.006) | 0.451 (0.006) | 0.347 (0.008) | 0.451 (0.006) | 0.344 (0.004) |
| | FDR | 0.679 (0.006) | 0.685 (0.006) | 0.657 (0.005) | 0.751 (0.007) | 0.657 (0.005) | 0.727 (0.004) |
| | MCC | 0.345 (0.005) | 0.351 (0.006) | 0.379 (0.005) | 0.258 (0.007) | 0.379 (0.005) | 0.296 (0.004) |
| MRS | | | | | | | |
| | TPR | 0.786 (0.008) | 0.799 (0.007) | **0.817 (0.008)** | **0.871 (0.010)** | **0.817 (0.008)** | 0.733 (0.007) |
| | FDR | 0.403 (0.010) | 0.438 (0.007) | 0.425 (0.007) | 0.268 (0.012) | 0.425 (0.007) | 0.561 (0.006) |
| | MCC | 0.678 (0.008) | 0.662 (0.007) | 0.678 (0.007) | 0.789 (0.012) | 0.678 (0.007) | 0.560 (0.006) |

Table 2: Average operating characteristics over 30 simulations for zero-inflated scenarios having $\sim 25\%$ zeros, $\sim 50\%$ zeros, and $\sim 75\%$ zeros, respectively. The standard error for each statistic is given in parentheses. The best performance is in boldface.

| Method | | Percentage of zeros | | |
| | | $\sim 25\%$ | $\sim 50\%$ | $\sim 75\%$ |
|---|---|---|---|---|
| ZIPBN | | | | |
| | TPR | **0.849 (0.010)** | **0.839 (0.010)** | **0.693 (0.010)** |
| | FDR | **0.230 (0.013)** | **0.180 (0.010)** | **0.312 (0.009)** |
| | MCC | **0.805 (0.012)** | **0.826 (0.010)** | **0.684 (0.010)** |
| ODS | | | | |
| | TPR | 0.370 (0.008) | 0.452 (0.006) | 0.317 (0.008) |
| | FDR | 0.648 (0.007) | 0.685 (0.006) | 0.780 (0.006) |
| | MCC | 0.348 (0.007) | 0.351 (0.006) | 0.246 (0.008) |
| MRS | | | | |
| | TPR | 0.776 (0.010) | 0.799 (0.007) | 0.681 (0.012) |
| | FDR | 0.403 (0.011) | 0.438 (0.007) | 0.805 (0.003) |
| | MCC | 0.673 (0.011) | 0.662 (0.007) | 0.343 (0.006) |

Table 3: Average operating characteristics over 30 simulations for misspecified models: Poisson BN and ZI-NegBin BN. The standard error for each statistic is given in parentheses. The best performance is in boldface and the second best is underlined. We did not apply BayPN to ZI-NegBin BN data, which is indicated by NA's.

| Method | Poisson BN | | | ZI-NegBin BN | | |
| | TPR | FDR | MCC | TPR | FDR | MCC |
|---|---|---|---|---|---|---|
| ZIPBN | 0.743 (0.012) | 0.259 (0.014) | 0.736 (0.013) | 0.698 (0.013) | **0.564 (0.013)** | **0.540 (0.013)** |
| BayPBN | **0.779 (0.008)** | **0.168 (0.010)** | **0.801 (0.009)** | NA | NA | NA |
| ODS | 0.569 (0.009) | 0.331 (0.009) | 0.609 (0.008) | 0.281 (0.008) | 0.832 (0.004) | 0.196 (0.005) |
| MRS | 0.517 (0.007) | 0.507 (0.006) | 0.494 (0.006) | **0.719 (0.010)** | 0.692 (0.006) | 0.455 (0.008) |

## 3.2 Real data analyses

We illustrated the proposed ZIPBN with two sets of analyses on a scRNA-seq dataset that consists of $\sim$30,000 intestinal cells from 5 aryl hydrocarbon receptor (AhR)-knockout mice.

### 3.2.1 Pairs of transcription factors and targets

We verified the identifiability of ZIPBN (Theorem 1) in real data. Specifically, we obtained a list of literature-curated pairs of transcription factor (X) and its target (Y) from the TRRUST database (Han *et al.*, 2018). This list provided biological ground truth of the cause-and-effect relationships, namely X→Y. For each pair in the list, we extracted the corresponding genes from our scRNA-seq dataset. Removing genes with more than 70% zeros, we retained 479 pairs for causal validation (the list is provided in the Supplementary Material). Subsequently, ZIPBN was applied to one pair at the time. Given the simplicity of the network space with two nodes, the sophisticated MCMC algorithm was not required. We were able to correctly identify 304 causal relationships out of 479 pairs. Compared to random guesses, the p-value was $1.1 \times 10^{-9}$ (binomial test with $H_0 : p = 0.5$ vs $H_a : p > 0.5$). For comparison, we applied MRS which correctly identified 198 causal relationships.

### 3.2.2 Pathway analysis

We expanded the analysis to $p = 40$ genes that are part of the Wnt signaling pathway. We focused on $n = 1,025$ cells from one AhR-knockout mouse. These cells were of the same cell type that was identified by the scRNA-seq clustering algorithm in R package `Seurat` (Butler *et al.*, 2018). We provide the list of genes and the data preprocessing procedure in the Supplementary Material. We ran $M = 10$ parallel chains for $5,000$ iterations with $2,500$ as burn-in and the same temperatures and prior specifications as in Section 3.1. The CPU time was 1.7 hours on an i9-9880H 2.3GHz CPU.

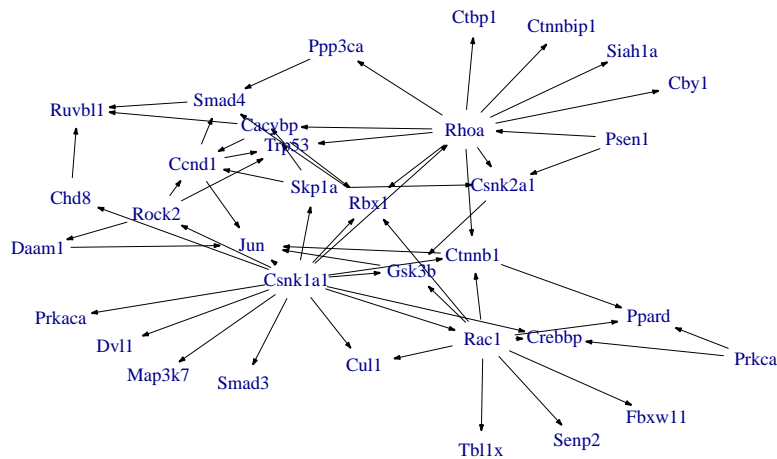

Figure 2: The estimated gene regulatory network using the ZIPBN models. Disconnected genes were not plotted: `Axin1`, `Csnk1e`, `Frat1`, `Mapk8`, `Myc`, and `Nlk`.

Figure 2 shows the estimated gene regulatory network using the posterior samples. The cutoff for determining $\hat{E}$ was chosen to control the posterior expected FDR at $1\%$. We found 60 edges in total, some of which are consistent with known gene regulations in the biological literature. For example, the estimated network confirmed gene regulation involving `Ctnnb1`, a key factor in the Wnt signaling pathway: `Ctnnb1` increases the transcription of `Jun` (Mann *et al.*, 1999) and `Csnk1a1` phosphorylates `Ctnnb1` (Amit *et al.*, 2002). Moreover, our network analysis also discovered several hub genes – `Csnk1a1`, `Rhoa`, and `Rac1`, with degrees of 15, 12, and 10, respectively. Hub genes are of particular importance because they are often involved in multiple regulatory activities. In fact, the importance of these genes has been well established. `Csnk1a1` is a core member of `Ctnnb1` destructive complex which interacts with multiple members of the Wnt signaling pathway (Amit *et al.*, 2002). `Rhoa` and `Rac1` regulate epithelial intercellular junctions via distinct morphological and biochemical mechanisms (Bruewer *et al.*, 2004).

## Broader Impact

The proposed ZIPBN will be useful for constructing causal gene regulatory network at the cell type level, which will assist biologists in generating causal hypotheses of gene regulation and expediting causal discovery processes. Without the proposed method, mechanistic understanding of cell-type-specific gene regulation will likely remain difficult. Additionally, the proposed method is broadly applicable to other applications (including educational psychology, ecology, behavior science, and economics) where data are zero-inflated and causal network inference is of interest.

## Acknowledgments and Disclosure of Funding

Ni's research was partially supported by the National Science Foundation (NSF DMS-1918851). Chapkin's research was partially supported by Texas AgriLife Research, the Sid Kyle Chair Endowment, the Allen Endowed Chair in Nutrition & Chronic Disease Prevention, the Cancer Prevention Research Institute of Texas (RP160589), and the National Institutes of Health (R01-ES025713, R01-CA202697, and R35-CA197707).

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
