[Supplementary Material]

# Supplementary Material of "Bayesian Causal Structural Learning with Zero-Inflated Poisson Bayesian Networks"

**Junsouk Choi**
Department of Statistics
Texas A&M University
College Station, TX 77843
jchoi@stat.tamu.edu

**Robert Chapkin**
Department of Nutrition
Texas A&M University
College Station, TX 77843
r-chapkin@tamu.edu

**Yang Ni**
Department of Statistics
Texas A&M University
College Station, TX 77843
yni@stat.tamu.edu

## A. Proof of Theorem 1

We provide a detailed proof for Theorem 1. Consider two Markov equivalent ZIPBNs $\mathcal{B} = (\mathcal{G}, \boldsymbol{\theta})$ and $\mathcal{B}' = (\mathcal{G}', \boldsymbol{\theta}')$ with $\mathcal{G} = (V, \boldsymbol{E})$, $\mathcal{G}' = (V, \boldsymbol{E}' \neq \boldsymbol{E})$, $\boldsymbol{\theta} = \{\boldsymbol{\alpha}, \boldsymbol{\beta}, \boldsymbol{\delta}, \boldsymbol{\gamma}\}$, and $\boldsymbol{\theta}' = \{\boldsymbol{\alpha}', \boldsymbol{\beta}', \boldsymbol{\delta}', \boldsymbol{\gamma}'\}$. Let $p(\boldsymbol{X}|\boldsymbol{E}, \boldsymbol{\theta})$ and $p(\boldsymbol{X}|\boldsymbol{E}', \boldsymbol{\theta}')$ denote the sampling distribution of $\boldsymbol{X}$ under $\mathcal{B}$ and $\mathcal{B}'$, respectively. We will prove by contradiction that $\mathcal{B}$ and $\mathcal{B}'$ are not distribution equivalent,

$$p(\boldsymbol{X}|\boldsymbol{E}, \boldsymbol{\theta}) \neq p(\boldsymbol{X}|\boldsymbol{E}', \boldsymbol{\theta}').$$

Suppose, on the contrary, that we can find a pair of $(\boldsymbol{\theta}, \boldsymbol{\theta}')$ such that $p(\boldsymbol{X}|\boldsymbol{E}, \boldsymbol{\theta}) = p(\boldsymbol{X}|\boldsymbol{E}', \boldsymbol{\theta}')$. Let $pa(j) = \{k \in V : e_{jk} = 1\}$ and $pa'(j) = \{k \in V : e'_{jk} = 1\}$ be the parent sets of node $j$ in $\mathcal{G}$ and $\mathcal{G}'$, respectively. DAG factorization leads to the following equation for all $x_1, \ldots, x_p$,

$$\sum_{j=1}^{p} \log Pr\left(X_j = x_j \,\big|\, \boldsymbol{X}_{pa(j)} = \boldsymbol{x}_{pa(j)}, \boldsymbol{E}, \boldsymbol{\theta}\right) = \sum_{j=1}^{p} \log Pr\left(X_j = x_j \,\big|\, \boldsymbol{X}_{pa'(j)} = \boldsymbol{x}_{pa'(j)}, \boldsymbol{E}', \boldsymbol{\theta}'\right). \tag{1}$$

Without loss of generality, we assume that the nodes are labeled such that there is no directed edge in $\boldsymbol{E}$ from later node to earlier node. Such labeling is also known as perfect/topological ordering of DAG $\mathcal{G}$. Define a set of edges that are connected to node $j$ and have opposite directions in $\boldsymbol{E}$ and $\boldsymbol{E}'$, $re(j) = \{k \in V : e_{jk} = e'_{kj} = 1\}$ for $j = 1, \ldots, p$. For $k \in re(j)$, $\boldsymbol{E}$ includes an edge $k \to j$, while $\boldsymbol{E}'$ has the reverse edge $j \to k$. If $re(j) = \emptyset$ for all $j$, there exists no pair of nodes $(j, k)$ such that $e_{jk} = e'_{kj} = 1$. This means $\boldsymbol{E} = \boldsymbol{E}'$, because Markov equivalent DAGs have the same skeleton. We will show by mathematical induction that $re(j) = \emptyset$ for all $j$, which contradicts the assumption that $\boldsymbol{E} \neq \boldsymbol{E}'$.

For node $p$ that is the last element of the perfect ordering of $\mathcal{G}$, we have $pa(p) = pa'(p) \cup re(p)$ due to the same skeleton of $\mathcal{G}$ and $\mathcal{G}'$. Taking the difference of the equality (1) at $(x_1, x_2, \ldots, x_p + 1)$ and $(x_1, x_2, \ldots, x_p)$ yields,

$$\sum_{k \in pa'(p) \cup re(p)} \beta_{pk} x_k + \gamma_p - \log(x_p + 1) = \sum_{k \in pa'(p)} \beta'_{pk} x_k + \gamma'_p - \log(x_p + 1) + \sum_{k \in re(p)} \Big[ \beta'_{kp} x_k -$$

$$\left\{ \exp\left( \beta'_{kp}(x_p + 1) + \sum_{l \in pa'(k) \setminus \{p\}} \beta'_{kl} x_\ell + \gamma'_k \right) - \exp\left( \beta'_{kp} x_p + \sum_{l \in pa'(k) \setminus \{p\}} \beta'_{kl} x_\ell + \gamma'_k \right) \right\} +$$

$$\log \frac{1 + \exp(\alpha'_{kp} x_p + \sum_{l \in pa'(k) \setminus \{p\}} \alpha'_{kl} x_\ell + \delta'_k)}{1 + \exp(\alpha'_{kp}(x_p + 1) + \sum_{l \in pa'(k) \setminus \{p\}} \alpha'_{kl} x_\ell + \delta'_k)} \Big],$$

for $x_1, \ldots, x_p > 0$ which can be simplified as,

$$\sum_{k \in pa'(p)} \beta_{pk} x_k + \gamma_p + \sum_{k \in re(p)} \left[ \beta_{pk} x_k + \exp\left( \beta'_{kp}(x_p + 1) + \sum_{l \in pa'(k) \setminus \{p\}} \beta'_{kl} x_\ell + \gamma'_k \right) \right] =$$

$$\sum_{k \in pa'(p)} \beta'_{pk} x_k + \gamma'_p + \sum_{k \in re(p)} \left[ \beta'_{kp} x_k + \exp\left( \beta'_{kp} x_p + \sum_{l \in pa'(k) \setminus \{p\}} \beta'_{kl} x_\ell + \gamma'_k \right) + \right.$$

$$\left. \log \frac{1 + \exp(\alpha'_{kp} x_p + \sum_{l \in pa'(k) \setminus \{p\}} \alpha'_{kl} x_\ell + \delta'_k)}{1 + \exp(\alpha'_{kp}(x_p + 1) + \sum_{l \in pa'(k) \setminus \{p\}} \alpha'_{kl} x_\ell + \delta'_k)} \right]. \tag{2}$$

The equation (2) holds for all $x_1, \ldots, x_p > 0$ if and only if $\beta_{pk} = \alpha'_{kp} = \beta'_{kp} = 0$ for $k \in re(p)$, $\beta_{pk} = \beta'_{pk}$ for $k \in pa'(p)$, and $\gamma_p = \gamma'_p$. The fact that $\alpha'_{kp} = \beta'_{kp} = 0$ for $k \in re(p)$ implies that $re(p) = \emptyset$. We also obtain that $\delta_p = \delta'_p$, $\gamma_p = \gamma'_p$, $\alpha_{pk} = \alpha'_{pk}$, $\beta_{pk} = \beta'_{pk}$ for $k \in pa(p) = pa'(p)$.

Now assume that for any $j = m+1, \ldots, p$, it holds that $re(j) = \emptyset$, $\delta_j = \delta'_j$, $\gamma_j = \gamma'_j$, $\alpha_{jk} = \alpha'_{jk}$, and $\beta_{jk} = \beta'_{jk}$ for $k \in pa(j) = pa'(j)$. We will show that it also holds for $j = m$. Define the child sets of node $m$ in graphs $\mathcal{G}$ and $\mathcal{G}'$ by $ch(m) = \{k \in V : e_{km} = 1\}$ and $ch'(m) = \{k \in V : e'_{km} = 1\}$, respectively. Again since $\mathcal{G}$ and $\mathcal{G}'$ share the same skeleton, $pa(m) = pa'(m) \cup re(m)$ and $ch'(m) = ch(m) \cup re(m)$. The difference of the equality (1) at $(x_1, \ldots, x_m + 1, \ldots, x_p)$ and $(x_1, \ldots, x_m, \ldots, x_p)$ is given by,

$$\sum_{k \in pa'(m) \cup re(m)} \beta_{mk} x_k + \gamma_m - \log(x_m + 1) + \sum_{k \in ch(m)} [\beta_{km} x_k -$$

$$\left\{ \exp\left( \beta_{km}(x_m + 1) + \sum_{l \in pa(k) \setminus \{m\}} \beta_{kl} x_\ell + \gamma_k \right) - \exp\left( \beta_{km} x_m + \sum_{l \in pa(k) \setminus \{m\}} \beta_{kl} x_\ell + \gamma_k \right) \right\} +$$

$$\log \frac{1 + \exp(\alpha_{km} x_m + \sum_{l \in pa(k) \setminus \{m\}} \alpha_{kl} x_\ell + \kappa_k)}{1 + \exp(\alpha_{km}(x_m + 1) + \sum_{l \in pa(k) \setminus \{m\}} \alpha_{kl} x_\ell + \gamma_k)} \bigg] =$$

$$\sum_{k \in pa'(m)} \beta'_{mk} x_k + \gamma'_m - \log(x_m + 1) + \sum_{k \in re(m) \cup ch(m)} [\beta'_{km} x_k -$$

$$\left\{ \exp\left( \beta'_{km}(x_m + 1) + \sum_{l \in pa'(k) \setminus \{m\}} \beta'_{kl} x_\ell + \gamma'_k \right) - \exp\left( \beta'_{km} x_m + \sum_{l \in pa'(k) \setminus \{m\}} \beta'_{kl} x_\ell + \gamma'_k \right) \right\} +$$

$$\log \frac{1 + \exp(\alpha'_{km} x_m + \sum_{l \in pa'(k) \setminus \{m\}} \alpha'_{kl} x_\ell + \delta'_k)}{1 + \exp(\alpha'_{km}(x_m + 1) + \sum_{l \in pa'(k) \setminus \{m\}} \alpha'_{kl} x_\ell + \delta'_k)} \bigg],$$

for all $x_1, \ldots, x_p > 0$. Note that $ch(m) \subset \{m+1, \ldots, p\}$. Hence, if $k \in ch(m)$, then we have $\delta_k = \delta'_k$, $\gamma_k = \gamma'_k$, $\alpha_{kl} = \alpha'_{kl}$, and $\beta_{kl} = \beta'_{kl}$ for $l \in pa(k) = pa'(k)$ due to the induction assumption. Simplifying the above equation, we obtain,

$$\sum_{k \in pa'(m)} \beta_{mk} x_k + \gamma_m + \sum_{k \in re(m)} \left[ \beta_{mk} x_k + \exp\left( \beta'_{km}(x_m + 1) + \sum_{l \in pa'(k) \setminus \{m\}} \beta'_{kl} x_\ell + \gamma'_k \right) \right] =$$

$$\sum_{k \in pa'(m)} \beta'_{mk} x_k + \gamma'_m + \sum_{k \in re(m)} \left[ \beta'_{km} x_k + \exp\left( \beta'_{km} x_m + \sum_{l \in pa'(k) \setminus \{m\}} \beta'_{kl} x_\ell + \gamma'_k \right) + \right.$$

$$\left. \log \frac{1 + \exp(\alpha'_{km} x_m + \sum_{l \in pa'(k) \setminus \{m\}} \alpha'_{kl} x_\ell + \delta'_k)}{1 + \exp(\alpha'_{km}(x_m + 1) + \sum_{l \in pa'(k) \setminus \{m\}} \alpha'_{kl} x_\ell + \delta'_k)} \right]. \tag{3}$$

The sufficient and necessary condition for the equation (3) to hold for all $x_1, \ldots, x_p > 0$ is that $\beta_{mk} = \alpha'_{km} = \beta'_{km} = 0$ for $k \in re(m)$, $\beta_{mk} = \beta'_{mk}$ for $k \in pa'(m)$, and $\gamma_m = \gamma'_m$. It follows that $re(m) = \emptyset$, and therefore we have $\delta_m = \delta'_m$, $\gamma_m = \gamma'_m$, $\alpha_{mk} = \alpha'_{mk}$, and $\beta_{mk} = \beta'_{mk}$ for $k \in pa(m) = pa'(m)$, which completes the proof.

## B. Alternative Proof of Identifiability of Poisson Bayesian Networks

We provide an alternative proof for identifiability of Poisson BN. For Poisson BNs, the conditional distribution of each node given its parents is assumed to be Poisson distribution,

$$Pr\left(X_j = x \mid \boldsymbol{X}_{pa(j)}\right) = \frac{\exp\left(-\lambda_j\right)\lambda_j^x}{x!},$$

where $\log(\lambda_j) = \sum_{k \in pa(j)} \beta_{jk} X_k + \gamma_j$. Here, Poisson BNs are parametrized by $\boldsymbol{\beta} = \{\beta_{jk}\}_{j,k}$, $\boldsymbol{\gamma} = \{\gamma_j\}_j$, and the network structure $\boldsymbol{E}$. Let $p\left(\boldsymbol{X}|\boldsymbol{E},\boldsymbol{\beta},\boldsymbol{\gamma}\right)$ and $p\left(\boldsymbol{X}|\boldsymbol{E}',\boldsymbol{\beta}',\boldsymbol{\gamma}'\right)$ denote the sampling distributions of two Markov equivalent Poisson BNs. As in the proof of Theorem 1, we will show by contradiction that

$$p\left(\boldsymbol{X}|\boldsymbol{E},\boldsymbol{\beta},\boldsymbol{\gamma}\right) \neq p\left(\boldsymbol{X}|\boldsymbol{E}',\boldsymbol{\beta}',\boldsymbol{\gamma}'\right).$$

Suppose that $p\left(\boldsymbol{X}|\boldsymbol{E},\boldsymbol{\beta},\boldsymbol{\gamma}\right) = p\left(\boldsymbol{X}|\boldsymbol{E}',\boldsymbol{\beta}',\boldsymbol{\gamma}'\right)$. Then it follows that

$$\sum_{j=1}^{p} \log Pr\left(X_j = x_j \mid \boldsymbol{X}_{pa(j)} = \boldsymbol{x}_{pa(j)}, \boldsymbol{E}, \boldsymbol{\beta}, \boldsymbol{\gamma}\right) = \sum_{j=1}^{p} \log Pr\left(X_j = x_j \mid \boldsymbol{X}_{pa'(j)} = \boldsymbol{x}_{pa'(j)}, \boldsymbol{E}', \boldsymbol{\beta}', \boldsymbol{\gamma}'\right),$$

(4)

for all $x_1, \ldots, x_p$. Without loss of generality, assume $(1, \ldots, p)$ to be the true causal ordering of $\boldsymbol{E}$. Recall that having $re(j) = \emptyset$ for all $j$ implies $\boldsymbol{E} = \boldsymbol{E}'$. We will show by mathematical induction that $re(j) = \emptyset$ for all $j$, which contradicts the assumption $\boldsymbol{E} \neq \boldsymbol{E}'$.

For node $p$, we obtain $pa(p) = pa'(p) \cup re(p)$ by combining two facts that (i) there is no directed edge in $\boldsymbol{E}$ pointing away from node $p$ and (ii) $\boldsymbol{E}$ has the same skeleton with $\boldsymbol{E}'$. The difference of the equality (4) at $(x_1, \ldots, x_p + 1)$ and $(x_1, \ldots, x_p)$ is calculated as

$$\sum_{k \in pa'(p) \cup re(p)} \beta_{pk} x_k + \gamma_p - \log(x_p + 1) = \sum_{k \in pa'(p)} \beta'_{pk} x_k + \gamma'_p - \log(x_p + 1) + \sum_{k \in re(p)} \Big[\beta'_{kp} x_k -$$

$$\left\{\exp\left(\beta'_{kp}(x_p + 1) + \sum_{l \in pa'(k)\backslash\{p\}} \beta'_{kl} x_\ell + \gamma'_k\right) - \exp\left(\beta'_{kp} x_p + \sum_{l \in pa'(k)\backslash\{p\}} \beta'_{kl} x_\ell + \gamma'_k\right)\right\}\Big].$$

It follows that

$$\sum_{k \in pa'(p)} \beta_{pk} x_k + \gamma_p + \sum_{k \in re(p)} \left[\beta_{pk} x_k + \exp\left(\beta'_{kp}(x_p + 1) + \sum_{l \in pa'(k)\backslash\{p\}} \beta'_{kl} x_\ell + \gamma'_k\right)\right]$$

$$= \sum_{k \in pa'(p)} \beta'_{pk} x_k + \gamma'_p + \sum_{k \in re(p)} \left[\beta'_{kp} x_k + \exp\left(\beta'_{kp} x_p + \sum_{l \in pa'(k)\backslash\{p\}} \beta'_{kl} x_\ell + \gamma'_k\right)\right] \quad (5)$$

for all $(x_1, \ldots, x_p)$. The equation (5) holds for all $(x_1, \ldots, x_p)$ if and only if $\beta_{pk} = \beta'_{kp} = 0$ for $k \in re(p)$, $\beta_{pk} = \beta'_{pk}$ for $k \in pa'(p)$, and $\gamma_p = \gamma'_p$. We can deduce $re(p) = \emptyset$ and $pa(p) = pa'(p)$ from the fact that $\beta_{pk} = \beta'_{kp} = 0$ for $k \in re(p)$.

Now assume that for any $j = m + 1, \ldots, p$, it holds that $re(j) = \emptyset$, $\gamma_j = \gamma'_j$, and $\beta_{jk} = \beta'_{jk}$ for $k \in pa(j) = pa'(j)$. We will show that it also holds for $j = m$. The same skeleton of $\boldsymbol{E}$ and $\boldsymbol{E}'$ yields $pa(m) \cup ch(m) = pa'(m) \cup re(m) \cup ch(m) = pa'(m) \cup ch'(m)$. Taking the difference of the equality (4) at $(x_1, \ldots, x_m + 1, \ldots, x_p)$ and $(x_1, \ldots, x_m, \ldots, x_p)$, we get,

$$\sum_{k \in pa'(m) \cup re(m)} \beta_{mk} x_k + \gamma_m - \log(x_m + 1) + \sum_{k \in ch(m)} \Big[\beta_{km} x_k -$$

$$\left\{\exp\left(\beta_{km}(x_m + 1) + \sum_{l \in pa(k)\backslash\{m\}} \beta_{kl} x_\ell + \gamma_k\right) - \exp\left(\beta_{km} x_m + \sum_{l \in pa(k)\backslash\{m\}} \beta_{kl} x_\ell + \gamma_k\right)\right\}\Big]$$

$$= \sum_{k \in pa'(m)} \beta'_{mk} x_k + \gamma'_m - \log(x_m + 1) + \sum_{k \in re(m) \cup ch(m)} \Big[\beta'_{km} x_k -$$

$$\left\{\exp\left(\beta'_{km}(x_m + 1) + \sum_{l \in pa'(k)\backslash\{m\}} \beta'_{kl} x_\ell + \gamma'_k\right) - \exp\left(\beta'_{km} x_m + \sum_{l \in pa'(k)\backslash\{m\}} \beta'_{kl} x_\ell + \gamma'_k\right)\right\}\Big],$$

for all $(x_1, \ldots, x_p)$. Since $ch(m) \subset \{m+1, \ldots, p\}$, we conclude that for all $k \in ch(m)$, $\gamma_k = \gamma'_k$ and $\beta_{kl} = \beta'_{kl}$ for $l \in pa(k) = pa'(k)$. Hence we can simplify the above equation as,

$$
\sum_{k \in pa'(m)} \beta_{mk} x_k + \gamma_m + \sum_{k \in re(m)} \left[ \beta_{mk} x_k + \exp\left( \beta'_{km}(x_m + 1) + \sum_{l \in pa'(k) \setminus \{m\}} \beta'_{kl} x_\ell + \gamma'_k \right) \right]
$$
$$
= \sum_{k \in pa'(m)} \beta'_{mk} x_k + \gamma'_m + \sum_{k \in re(m)} \left[ \beta'_{km} x_k + \exp\left( \beta'_{km} x_m + \sum_{l \in pa'(k) \setminus \{m\}} \beta'_{kl} x_\ell + \gamma'_k \right) \right].
$$
(6)

The equation (6) holds for all $(x_1, \ldots, x_p)$ if and only if $\beta_{mk} = \beta'_{km} = 0$ for $k \in re(m)$, $\beta_{mk} = \beta'_{mk}$ for $k \in pa'(m)$, and $\gamma_m = \gamma'_m$. Therefore, we obtain $re(m) = \emptyset$ and $pa(m) = pa'(m)$, which completes the proof. Finally, we remark that in Park & Raskutti (2015), their proof exploited the overdispersion properties of Poisson BNs whereas our proof does not rely on that specific property and is therefore generalizable to other distribution families such as negative binomial and continuous distributions (results not shown).

## C. Analytical Integration of Equations (3)-(5)

We integrate out $\rho$ from $p(\boldsymbol{E}, \rho)$,

$$
p(\boldsymbol{E}) = \int p(\boldsymbol{E}|\rho) p(\rho) d\rho
$$
$$
\propto \int z(\rho)^{-1} \prod_{j \neq k} \rho^{e_{jk}} (1-\rho)^{1-e_{jk}} I(\mathcal{G} \in \mathcal{D}) z(\rho) \rho^{a_\rho - 1} (1-\rho)^{b_\rho - 1} d\rho
$$
$$
= I(\mathcal{G} \in \mathcal{D}) \int \rho^{\sum_{j \neq k} e_{jk} + a_\rho - 1} (1-\rho)^{\sum_{j \neq k}(1 - e_{jk}) + b_\rho - 1} d\rho
$$
$$
= B\left( \sum_{j \neq k} e_{jk} + a_\rho, \sum_{j \neq k}(1 - e_{jk}) + b_\rho \right) I(\mathcal{G} \in \mathcal{D}),
$$

where the last equality holds because the integrand is the kernel of a beta distribution.

## D. More Details on the Real Data Analyses

**Collection Process of AhR-Knockout scRNA-seq Data.** The scRNA-seq experiments were performed on five mice with AhR knockout targeted to intestinal stem cells. AhR is a ligand-activated transcription factor that is capable of integrating external environmental stimuli and host responses to modulate intestinal stem cell development, tissue regeneration, and colon cancer risk (Kim *et al.*, 2016; Safe *et al.*, 2018). We use this dataset to construct gene regulatory networks for genes that belong to Wnt signaling pathway. On average each mouse contributed ∼6,000 cells.

**List of Pairs of Transcription Factors and Targets.** We provide the complete list of pairs of transcription factors and targets used in Section 3.2.1 in Tables T1 and T2.

**List of Genes in Wnt Pathway.** We have used following genes in Section 3.2.2: Axin1, Cby1, Ccnd1, Chd8, Cacybp, Crebbp, Csnk1a1, Csnk1e, Csnk2a1, Ctbp1, Ctnnbip1, Ctnnb1, Cul1, Daam1, Dvl1, Frat1, Fbxw11, Gsk3b, Jun, Mapk8, Map3k7, Myc, Nlk, Senp2, Ppard, Ppp3ca, Prkaca, Prkca, Psen1, Rac1, Rbx1, Rhoa, Rock2, Ruvbl1, Siah1a, Skp1a, Smad3, Smad4, Tbl1x, and Trp53.

**Data Preprocessing using Seurat Package.** Cells of the same type were used to recover the gene regulatory network in Section 3.2.1. To identify the cell type, we clustered data from one mouse by using the scRNA-seq clustering algorithm in R package `Seurat` (Butler *et al.*, 2018). Before applying the algorithm, we normalized the data with the LogNormalize method and selected $2,000$ genes that showed high cell-to-cell variation. The principal component analysis was done on the scaled data to

calculate the distance between cells for the clustering analysis. We considered the first 19 principal components to build the cellular distance matrix. The clustering algorithm in `Seurat` was applied to construct a K-nearest neighborhood graph based on the distance matrix and partition it into several clusters. The resolution parameter of $0.1$ was specified, resulting in 9 cell types in total. We chose the cell type that includes $n = 1,025$ cells with $60\%$ zeros, which is comparable to our simulation study in Section 3.1.

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

T1. Pairs of Transcription Factors (TF) and Targets (TG).

| TF | TG | TF | TG | TF | TG | TF | TG | TF | TG |
|---|---|---|---|---|---|---|---|---|---|
| Aes | Ihh | Egr1 | Ccnd2 | Fosb | Cdh1 | Jun | Odc1 | Nfkb1 | Akt1 |
| Apc | Ctnnb1 | Egr1 | Crebbp | Fosb | Fos | Jun | Plin2 | Nfkb1 | Xiap |
| Apc | Tcf4 | Egr1 | Eno1 | Foxa1 | Cbx5 | Jun | Nectin2 | Nfkb1 | Xiap |
| Apc | Trp53 | Egr1 | Hsd11b2 | Foxa1 | Muc2 | Jun | Rab18 | Nfkb1 | Ccnd1 |
| Apex1 | Nfkb1 | Egr1 | Id3 | Foxa3 | Cdx2 | Jun | Serpinb5 | Nfkb1 | Ccnd1 |
| Apex1 | Nfkb1 | Egr1 | Ifngr1 | Fus | Sod2 | Jun | Tpr | Nfkb1 | Cd38 |
| Arid5b | Sox9 | Egr1 | Jun | Gata6 | Cdx2 | Jun | Trp53 | Nfkb1 | Cdk4 |
| Atf4 | Hspa5 | Egr1 | Jund | Gtf2i | Ccnd1 | Jun | Trp53 | Nfkb1 | Ctnnb1 |
| Atf4 | Ihh | Egr1 | Map1lc3b | Gtf2i | Cfdp1 | Junb | Ccnd1 | Nfkb1 | Cxadr |
| Cdx1 | Cdx2 | Egr1 | Ppp1r1b | Gtf2i | Csrp2 | Jund | Jun | Nfkb1 | Ebna1bp2 |
| Cdx2 | Cdh17 | Egr1 | Pten | Gtf2i | Fos | Keap1 | Nfe2l2 | Nfkb1 | Elf3 |
| Cdx2 | Cdx1 | Egr1 | Pten | Gtf2i | Fos | Khdrbs1 | Ece1 | Nfkb1 | Kdm2a |
| Cdx2 | Glb1 | Egr1 | Ptges2 | Gtf2i | Hspa5 | Klf4 | Ccnd2 | Nfkb1 | Fos |
| Cdx2 | Kitl | Egr1 | Smad7 | Gtf2i | Id1 | Klf4 | Cdh1 | Nfkb1 | Fth1 |
| Cdx2 | Mgam | Egr1 | Stim1 | Gtf2i | Nsd1 | Klf5 | Egr1 | Nfkb1 | Gclm |
| Cebpz | Hspa1b | Egr1 | Trp53 | Hdac1 | Jun | Klf6 | Asah1 | Nfkb1 | Glrx |
| Chd4 | Jun | Ehmt2 | Ccnd1 | Hdac1 | Nfkbia | Klf6 | Pttg1 | Nfkb1 | Hes1 |
| Chd7 | Sox4 | Ehmt2 | Pparg | Hdac1 | Pparg | Mapk1 | Egr1 | Nfkb1 | Hmgcr |
| Chd8 | Stat3 | Ep300 | Ccnd1 | Hdac1 | Ppp2ca | Max | Bax | Nfkb1 | Hsd11b2 |
| Chd8 | Tcf4 | Ep300 | Ccnd1 | Hdac1 | Sp1 | Max | Odc1 | Nfkb1 | Hsd11b2 |
| Cited2 | Pparg | Ep300 | Clock | Hdac1 | Epcam | Max | Trp53 | Nfkb1 | Junb |
| Clock | Bhlhe40 | Ep300 | Fasn | Hdac2 | Nfkbia | Mdm4 | Trp53 | Nfkb1 | Klf5 |
| Clock | Bhlhe40 | Ep300 | Fos | Hdac2 | Smarca4 | Mdm4 | Trp53 | Nfkb1 | Mbp |
| Creb3 | Herpud1 | Ep300 | Irf2 | Hdac3 | Fos | Med1 | Pparg | Nfkb1 | Myb |
| Crebbp | Clock | Ep300 | Klf5 | Hdac3 | Jun | Med1 | Pparg | Nfkb1 | Nfkbia |
| Crebbp | Ep300 | Ep300 | Ldha | Hdac3 | Nfkbia | Myb | Gstm2 | Nfkb1 | Pla2g4a |
| Crebbp | Fos | Ep300 | Mt1 | Hes6 | Hes1 | Myb | H2afz | Nfkb1 | Plin2 |
| Crebbp | Fosb | Ep300 | Myb | Hnf4a | Cdx2 | Myb | Ppm1a | Nfkb1 | Psme2 |
| Crebbp | Fth1 | Ep300 | Pparg | Hnf4a | Foxa1 | Nab1 | Egr1 | Nfkb1 | Pten |
| Crebbp | Ldha | Ep300 | Tapbp | Hnf4a | Gstp1 | Nab1 | Ifngr1 | Nfkb1 | Sod2 |
| Crebbp | Med1 | Ep300 | Txnip | Hnf4a | Perp | Ncor1 | Hdac3 | Nfkb1 | Sox9 |
| Crebbp | Mt1 | Esrra | Acadm | Id3 | Smarcc1 | Ncor1 | Hes1 | Nfkb1 | Tapbp |
| Crebbp | Nfkb1 | Esrra | Nrip1 | Irf3 | Pura | Ncor1 | Nfkbia | Nfkb1 | Trp53 |
| Crebbp | Trp53 | Esrra | Rb1cc1 | Irf3 | Socs2 | Nfat5 | Akr1b3 | Nfkb1 | Ube2h |
| Crebbp | Trp53 | Esrra | Trp53 | Irf8 | Jak1 | Nfat5 | Bax | Nfkb1 | Yy1 |
| Ctnnb1 | Casp3 | Ets2 | Egr1 | Jun | Asah1 | Nfe2l2 | Akr1b3 | Nfkb1 | Ccnd1 |
| Ctnnb1 | Ccnd1 | Ets2 | Hmox2 | Jun | B4galnt1 | Nfe2l2 | Cox17 | Nfkb1 | Trp53 |
| Ctnnb1 | Ccnd1 | Ets2 | Jun | Jun | Cat | Nfe2l2 | Fos | Nfkb1 | Cdh1 |
| Ctnnb1 | Cdh1 | Ets2 | Krt8 | Jun | Ccnd1 | Nfe2l2 | Gclc | Nfkbia | Nfkb1 |
| Ctnnb1 | Cdx1 | Ezh2 | Bad | Jun | Ccnd1 | Nfe2l2 | Keap1 | Nr1h2 | Fasn |
| Ctnnb1 | Cdx2 | Ezh2 | Creb3l1 | Jun | Ccnd1 | Nfe2l2 | Keap1 | Nr3c1 | Cdk4 |
| Ctnnb1 | Fgfbp1 | Ezh2 | Trp53 | Jun | Ccnd2 | Nfe2l2 | Mgst1 | Nr3c1 | Cdk6 |
| Ctnnb1 | Glul | Fos | Acp5 | Jun | Fos | Nfe2l2 | Mt1 | Nr3c1 | Hes1 |
| Ctnnb1 | Ihh | Fos | Egr1 | Jun | Fos | Nfe2l2 | Nfkb1 | Nr3c1 | Nfkbia |
| Ctnnb1 | Ppard | Fos | H2-K1 | Jun | Gclc | Nfe2l2 | Pparg | Nr3c1 | Vdr |
| Ctnnb1 | Smc3 | Fos | Hmgcr | Jun | Hmgcr | Nfe2l2 | Pparg | Nr5a2 | Ccnd1 |
| Ctnnb1 | Tcf4 | Fos | Jun | Jun | Krt18 | Nfe2l2 | Prdx1 | Nr5a2 | Kras |
| Cux1 | Dctn6 | Fos | Krt18 | Jun | Nfe2l2 | Nfe2l2 | Prdx1 | Nupr1 | Ep300 |
| Ddx54 | Mbp | Fos | Mt1 | Jun | Nfe2l2 | Nfe2l2 | Prdx2 | Pbx1 | Cdh1 |
| Dnajc2 | Id1 | Fos | Nfe2l2 | Jun | Nfkb1 | Nfe2l2 | Sdha | Pcbp2 | Trp53 |
| Egr1 | Bax | Fos | Nr3c1 | Jun | Nfkb1 | Nfe2l2 | Sod2 | Pnn | Cdx2 |
| Egr1 | Cat | Fos | Stim1 | Jun | Nr3c1 | Nfe2l2 | Txnip | Ppard | Insig1 |
| Egr1 | Ccnd1 | Fos | Trp53 | Jun | Nr3c1 | Nfe2l2 | Txnrd1 | Ppard | Pparg |

T2. Pairs of Transcription Factors (TF) and Targets (TG).

| TF | TG | TF | TG | TF | TG | TF | TG | TF | TG |
|---|---|---|---|---|---|---|---|---|---|
| Ppard | Sp1 | Sp1 | Cops5 | Sp3 | Rest | Trp53 | Epcam | Zbtb7a | Creb3 |
| Pparg | Atp2a2 | Sp1 | Cox17 | Sp3 | Sdc1 | Trp53 | Fos | Zfp36 | Nfkb1 |
| Pparg | Cat | Sp1 | Cox4i1 | Sp3 | Sp1 | Trp53 | Fos | | |
| Pparg | Ccnd1 | Sp1 | Csrp2 | Sp3 | Sp1 | Trp53 | Gsk3b | | |
| Pparg | Cops5 | Sp1 | Cxadr | Sp3 | Tspo | Trp53 | Hdgf | | |
| Pparg | Dbi | Sp1 | Egr1 | Sp3 | Ucp2 | Trp53 | Hras | | |
| Pparg | Egr1 | Sp1 | Esrra | Srebf1 | Acss2 | Trp53 | Hras | | |
| Pparg | Insr | Sp1 | Fos | Srebf1 | Fasn | Trp53 | Hspa1b | | |
| Pparg | Klf5 | Sp1 | Gstk1 | Srebf1 | Fasn | Trp53 | Insr | | |
| Pparg | Plin2 | Sp1 | Hmgcl | Srebf1 | Hmgcr | Trp53 | Krt19 | | |
| Pparg | Sod2 | Sp1 | Hmgcr | Srebf2 | Acss2 | Trp53 | Mapk1 | | |
| Pparg | Trp53 | Sp1 | Hspa1b | Srebf2 | Hmgcr | Trp53 | Mcts1 | | |
| Pparg | Ucp2 | Sp1 | Hspa1b | Srebf2 | Hmgcs1 | Trp53 | Mt1 | | |
| Ppargc1b | Esrra | Sp1 | Insr | Srebf2 | Hnf4a | Trp53 | Nfkb1 | | |
| Pttg1 | Pttg1ip | Sp1 | Itga6 | Stat3 | Akt1 | Trp53 | Nr3c1 | | |
| Pura | Mbp | Sp1 | Jun | Stat3 | Ccnd1 | Trp53 | Nupr1 | | |
| Raf1 | Mapk1 | Sp1 | Jund | Stat3 | Ccnd2 | Trp53 | Pcbp2 | | |
| Rala | Fos | Sp1 | Kdm5a | Stat3 | Egr1 | Trp53 | Perp | | |
| Rest | Kcnn4 | Sp1 | Lmna | Stat3 | Fos | Trp53 | Pla2g16 | | |
| Satb2 | Upf3b | Sp1 | Mapk1 | Stat3 | Fos | Trp53 | Prnp | | |
| Sf1 | Sox9 | Sp1 | Mbp | Stat3 | Klf4 | Trp53 | Psen1 | | |
| Smad1 | Id3 | Sp1 | Mgst1 | Stat3 | Mt1 | Trp53 | Psme3 | | |
| Smad4 | Cdh1 | Sp1 | Mif | Stat3 | Phb | Trp53 | Pten | | |
| Smad4 | Cldn3 | Sp1 | Myb | Stat3 | Pik3r1 | Trp53 | Rchy1 | | |
| Smad4 | Cxadr | Sp1 | Myb | Stat3 | Pnp | Trp53 | Reep5 | | |
| Smad4 | Id3 | Sp1 | Nr1h2 | Stat3 | Pten | Trp53 | Rps6ka1 | | |
| Smad4 | Jun | Sp1 | Pla2g16 | Stat3 | Srebf1 | Trp53 | Serpinb5 | | |
| Smad4 | Ocln | Sp1 | Ppm1a | Stat3 | Tcf4 | Trp53 | Siva1 | | |
| Smad4 | Pparg | Sp1 | Prdx6 | Stat3 | Trp53 | Trp53 | Slc6a6 | | |
| Smad4 | Smad7 | Sp1 | Prkra | Suz12 | Cdh1 | Trp53 | Sod2 | | |
| Smad7 | Glb1 | Sp1 | Ptges2 | Tardbp | Ran | Trp53 | Srebf1 | | |
| Smad7 | Nfkbia | Sp1 | Nectin2 | Tcf12 | Smarcc1 | Trp53 | Tcf4 | | |
| Smarca4 | Calm1 | Sp1 | Rest | Tcf4 | Cdh1 | Trp53 | Tcf7l2 | | |
| Smarca4 | Cdh1 | Sp1 | Slc3a2 | Tcf4 | Cdx2 | Trp53 | Tpt1 | | |
| Smarca4 | Nfkbia | Sp1 | Slc5a1 | Tcf4 | Cops5 | Trp53 | Tpt1 | | |
| Smarca4 | Rest | Sp1 | Slc5a1 | Tcf4 | Id3 | Trp53 | Tyms | | |
| Sox4 | Ctnnb1 | Sp1 | Smad7 | Tcf4 | Smc3 | Trp53 | Klf4 | | |
| Sox4 | Ezh2 | Sp1 | Smarcc1 | Tcf7l2 | Cdx1 | Ube2k | Nfe2l2 | | |
| Sox4 | Mecom | Sp1 | Sod2 | Thrap3 | Pparg | Vdr | Ctnnb1 | | |
| Sox4 | Tcf4 | Sp1 | Srebf1 | Tob1 | Ccnd1 | Vezf1 | Morf4l1 | | |
| Sox9 | Cdh1 | Sp1 | Srebf1 | Trim28 | Sox9 | Vezf1 | Stmn1 | | |
| Sox9 | Ctnnb1 | Sp1 | Sult1a1 | Trp53 | Bak1 | Xbp1 | Fasn | | |
| Sp1 | Abcb1a | Sp1 | Tpt1 | Trp53 | Bax | Xbp1 | Herpud1 | | |
| Sp1 | App | Sp1 | Tpt1 | Trp53 | Bax | Xbp1 | Tmbim6 | | |
| Sp1 | Asah1 | Sp1 | Tspo | Trp53 | Btg2 | Ybx1 | Abcb1a | | |
| Sp1 | Bhlhe40 | Sp1 | Tyms | Trp53 | Casp3 | Ybx1 | Nono | | |
| Sp1 | Bsg | Sp1 | Ucp2 | Trp53 | Casp6 | Yy1 | Cdh1 | | |
| Sp1 | Ccnd1 | Sp3 | Bsg | Trp53 | Casp7 | Yy1 | Fos | | |
| Sp1 | Ccnd1 | Sp3 | Itga6 | Trp53 | Casp8 | Yy1 | Hspa5 | | |
| Sp1 | Ccnd2 | Sp3 | Lmna | Trp53 | Ddit4 | Yy1 | Nr3c1 | | |
| Sp1 | Cdk6 | Sp3 | Mapk1 | Trp53 | Ddit4 | Yy1 | Tcf4 | | |
| Sp1 | Ces1d | Sp3 | Pla2g16 | Trp53 | Egr1 | Yy1 | Trp53 | | |
| Sp1 | Cirbp | Sp3 | Prkra | Trp53 | Ei24 | Zbtb20 | Sox9 | | |