[Reviews · NeurIPS 2020]

Review 1

Summary and Contributions: This paper proposes a zero-inflated Poisson Bayesian networks for modelling zero-inflated count data. Two main contributions are (i) proving that the model is identifiable with cross-sectional data; (ii) providing a fully Bayesian inference algorithm for causal discovery from ZIPBN models; and (iii) evaluating the new ZIPBN model for different sample size and node size combinations. Priors are placed on the edge set and the associated distributional parameters. Causal discovery is carried out via parallel-tempered MCMC that allows for cross-chain state swapping and Gibbs sampling. The Gibbs sampler permits edge additions or removals, and edge reversals. The new ZIPBN outperformed other structure learning methods for discrete BNs and was demonstrated on 2 real world data sets.

Strengths: This paper addresses a practical problem that often plagues population size estimation from count data. The proof of identifiability does not hinge on presence overdispersion but relies on showing that Markov equivalent BNs are not distribution equivalent It is demonstrated that the new method is robust to the zero-inflation property when there is approximately 50% zeros in the data. The ZIPBN model still outperforms other models when there is no zero-inflation case and the no zero-inflation case can easily be fit via ZIPBN by setting a tuning parameter to zero.

Weaknesses: The authors considered two scenarios in simulations: (i) number nodes is constant but sample size grows; and (ii) the sample size is fixed but the number of nodes grows. Under both settings, the same amount of zero-inflation is present (~50%). However, it is often the case that as the number of nodes grows, the higher the amount of zero-inflation there may be in the data. Comparing the p=25 and p=75 settings, is it reasonable for both to have about 50% zeroes in the observations (or might there me more zeros in the larger node setting?). I would imagine this problem would get worse in the setting of causal learning from scRNA-seq data. No comment on computational complexity or how the algorithm scales with either sample size or node size is provided.

Correctness: The main identifiability claim seems sound but see comment on weaknesses above.

Clarity: Given the amount of technical details required to present the MCMC procedure, the authors have done a good job. However, since a main contribution is the proof of identifiability, I wish they had expanded to provide more intuition into why distributional equivalence does not hold, at least in the base case. Although code is provided as supplementary material, the authors should state what p_s was in the experiments and how it was determined. Line 287: Of the 60 edges, how many were consistent with known gene regulation? Q. How long did it take to analyze the scRNA-seq data?

Relation to Prior Work: There appear to be two main alternatives to ZIPBN, though neither was formulated to handle zero-inflation. The authors take a fully Bayesian approach for inference, so the work here is not a trivial extension of this prior work.

Reproducibility: Yes

Additional Feedback: Minor comments: When presenting numbers in the thousands, remove the space after the comma (see lines 234 and 248 for examples). Line 1 of Algorithm 1: Shouldn't p_s = probability of a swap update be listed as an input? And if not, then details are needed for how p_s is determined. Post-rebuttal: After reading the authors' rebuttal I am satisfied that they have addressed most of the concerns raised by the reviewers. I feel they have the space to add the needed details and trust they will do this. There is a slight concern about sending it to press without seeing it one more time. However, the work is a solid contribution to the NeurIPS community.


Review 2

Summary and Contributions: The paper develops a casual model for modelling zero-inflated count data and proposes a MCMC method for Bayesian learning of the model. The authors show through synthetic experiments that their model is better able to model causal relationships between zero-inflated count data as compared to previous approaches. The authors also validate their proposed approach through some limited experiments on real world data. After rebuttal: With the additional results the authors have addressed my concerns about misspecified models. I think the paper makes a strong case for developing a new Bayesian network model for modeling zero-inflated count data.

Strengths: 1. The problem is well-motivated and there exists cases where one would like to learn Bayesian networks for zero-inflated data. The work builds upon the earlier work of Park and Raskutti (2015) and Park and Park (2019) that develop various causal models for count data, by extending them to zero-inflated count data. 2. The proposed model is identifiable which ensures that causal effects can be recovered from purely observational data. 3. The proposed MCMC algorithm for Bayesian learning of the model is sound and well designed.

Weaknesses: 1. Given that the authors propose a new model for zero-inflated count data, more thorough real-world experiments were needed to prove the usefulness of the model. For instance it is not clear, whether the edges recovered are causal. This would be clear by comparing against edges returned by simply performing zero-inflated one-vs-rest Poisson regression. 2. Zero-inflated count data can also be modeled by the negative Binomial distribution if the number of zeros aren't too many. Therefore this can be modeled by BNs for the hypergeometric family developed by Park and Park (2019). In the synthetic experiments, the authors generate data according to the generative model (i.e. generated by a ZIPBN) , therefore the proposed method out-performs the MRS algorithm. The performance of the method vis-a-vis MRS when the number of zeros is smaller and when the zero-inflated data doesn't follow the data generating process would give a better idea of the performance of the method. 3. Scalability of the proposed method is not discussed. It is not clear what the running time of the MCMC algorithm is on the size of graphs considered in the paper and if the method can scale to a large number of variables. The real-world experiments only consider a graph over 40 nodes.

Correctness: The identifiability results seem to be correct and the proposed MCMC formulation is sound.

Clarity: Yes.

Relation to Prior Work: The paper is well-positioned with respect to prior work and extends a line of work on modeling count data to zero-inflated count data.

Reproducibility: No

Additional Feedback: Some of the real-world data used in experiments is not publicly available.


Review 3

Summary and Contributions: This paper introduces an algorithm, ZIPBN, for learning the structure of a Bayesian network (BN) when the sampling distribution is a zero-inflated Poisson (ZIP) distribution. A ZIP distribution models counts using one component that models all zero counts and another that models all possible counts. The paper provides a theorem and proof that in the large sample limit ZIPBN can distinguish ZIP BNs that are Markov equivalent in that they represent the same d-separation relationships. The ZIPBN algorithm is evaluated using simulated and real data. The real data include single-cell RNA-seq data that were used to evaluate whether ZIPBN could learn a well-known cellular pathway.

Strengths: There are particular situations in which it is reasonable to model all the variables in a dataset as having a zero-inflated Poisson distribution and then be able to learn a BN structure on those variables. It is useful to have a Bayesian algorithm that is able to learn BN structure under such circumstances. The evaluations using simulated and real data provide some support for that the ZIPBN works well, although see below for some concerns.

Weaknesses: The paper emphasizes its focus on causal structure learning. In doing so it assumes “causal sufficiency”, that is, it assumes that there are no latent confounders of the measured variables. Generally, there are many latent confounders of the measured variables in most domains. In the past 20+ years, there has been substantial progress in developing graphical representations and algorithms for learning equivalence classes of causal networks from observational data. When causal sufficiency is assumed, the learning of DAG structure is generally called Bayesian network structure learning, not causal structural learning, as in the title of the paper. It would be helpful for the paper to more prominently highlight this assumption. The paper states that “unlike heuristic/greedy search algorithms, MCMC is theoretically guaranteed to converge to its stationary distribution.” However, there exist greedy algorithms for learning BNs that are guaranteed to converge to the data generating BN structure in the large sample limit under reasonable assumptions. See: Chickering DM. Optimal structure identification with greedy search. Journal of Machine Learning Research. 2002;3(Nov):507-54. In the evaluation using simulated data, it is difficult to know the reason for the superior performance of the ZIPBN method, because it differs in several ways from the other applied methods. It would be helpful if the ZIPBN method had been modified to reduce the number of differences in order to isolate the reasons for those differences. For example, by using a uniform prior over all BN structures in ZIPBN and applying it also to the simulated data, we would have a better understand the roll of the structure prior in driving ZIPBN to perform better than the other methods. There are some weaknesses in the evaluation of the real data. In the study involving pairs, presumably the goal was to determine for each studied pair which one of the following edge types is true: A  B, A  B, A … B, where “…” means no edge. It would be useful to state this goal explicitly. The filtering of the pairs before learning may have made the task of identifying the true relationships easier. It would be helpful to use unfiltered pairs during learning and then evaluate just those pairs for which the true causal relationship is known. In the study involving pathway analysis, what is the cutoff used in controlling the FDR and how was it determined? Also, it seems important to report an overall performance statistic for the ZIPBN algorithm and compare it to the performance of other methods, such as those used in the simulation study. Without such performance statistics and comparisons, it is difficult to assess how ZIPBN performed on this real data. In all the evaluation studies, it would be useful to report the running times of all the algorithms.

Correctness: The conceptual basis of the ZIPBN appears sound, although there are some details that are not clear to me (see below). The empirical work is informative, although the previous section describes some concerns. I have the following comments about the theoretical work (Theorem 1): * Line 5.5: It would be helpful to add a universal or existential quantifier on theta and theta’ to the equation. Also, this equation is with regard to the overall joint probability distributions of the two models. Proving the inequality would not seem to prove that all Markov equivalences can be resolved, but only that some of them can be. * Line 16: It would be useful if the proof were more explicit in pointing out where the assumption that E <> E’ is made in the text that appears above this line.

Clarity: Overall, the paper is grammatically well written. Some parts are difficult to follow, as summarized below. For the statement of Theorem 1, it would be useful to first formally define the particular meaning of “identifiable” being used in the paper. The section on “Prior of E” is not very clear. Please show the integral over the product of Equations 3 and 4, or at least mention it as being what is solved to derive Equation 5. Also, integrating over rho is not necessary to obtain multiplicity control in structure learning, although the paper seems to suggest it is. The section on “Prior of Theta” also is not very clear in several places, including the discussion of intercepts. The T_i terms are introduced before being defined. Section 2.4: It would be helpful to give a simple example of the steps in this section, either in the main paper or in the supplement.

Relation to Prior Work: The paper discusses well previous work that is most closely related to the ZIPBN algorithm. It does less well in discussing the broader set of work done in causal structure learning, including past work on Bayesian methods. It also does not discuss the issue of latent confounders, the importance of this issue in causal discovery, and previous work on developing methods for learning causal models that represent latent confounding.

Reproducibility: No

Additional Feedback: The real data are stated to be not yet published, and thus, not yet available for other researchers to use. I also did not see a pointer to a location where the simulated data are available. >>> Comments after the author feedback: The authors did a very good job in responding to the reviewer critiques. They addressed many of the issues in some detail. I no longer have a concern about the proof of Theorem 1. Due to space limitations in providing feedback, for other issues, such as clarifying the description of priors, they indicate that they plan to address them in revising the paper.

[Author Response · NeurIPS 2020]

We thank reviewers for their thoughtful comments and provide responses to, in our opinion, the most prominent ones.
**[R1, R2] Zero inflation and model misspecification.** We now tested ZIPBN on different percentages of zeros (0%,
25%, 50%, 75%) using otherwise the same setting as in Tab. 2. ZIPBN outperformed MRS, e.g., MCCs were 0.74
vs 0.49, 0.80 vs 0.67, 0.83 vs 0.66, and 0.69 vs 0.34. We also tested ZIPBN on misspecified models, namely, zero-
inflated negative binomial BN with dispersion parameters generated uniformly in $(1, 5)$ and 50% zeros. ZIPBN still
outperformed MRS with a smaller gap (MCC 0.54 vs 0.46). We anticipate MRS to outperform ZIPBN as %zeros$\rightarrow$0.
**[All] Real data validation**. It is difficult to directly quantify and compare the estimation accuracy for real large
networks due to unknown true structure, which motivated our first real data analysis with transcription factors. For
comparison, we now applied the closest competitor MRS to the same dataset, which correctly identified 198 pairs (we
identified 304) out of 479. For the pathway analysis, we now computed AIC for ZIPBN and MRS. Because MRS only
provides an estimate of the graph, we fit a zero-inflated Poisson regression to each node given its parents estimated
from MRS. ZIPBN outperformed MRS with AIC 84840.13 vs 85142.73 (the two graphs have similar sparsity).
**[R3] Data filtering.** We agree that filtering out genes with >70% zeros makes the task easier. We did so mainly because
genes with >70% zeros also tend to have very low non-zero counts and therefore have extremely small variability across
samples (e.g. in our study, the median variance of genes with >70% zeros is $10^{-4}$). For that reason, removing those
uniformly low expressed genes seems to be a common practice in genomic studies. Note that with the filtering, the
remaining genes still have about 50% zeros and low variability (median variance is 1.62), for which our method worked
better than competing methods (see the previous paragraph). The purpose of this study is not to prove that ZIPBN
works well in extremely sparse data but rather to provide real-world evidence in addition to simulations that when the
zero-inflation is moderately large (<70%), ZIPBN is capable of and superior in identifying the correct DAG.
**[R3] Causal sufficiency.** Causal sufficiency should have been stated explicitly. We are also aware that papers addressing
latent confounders (e.g., Spirtes and Richardson 2002, Salehkaleybar et al. 2020) have been extensively studied. We had
focused on the literature most closely related to ours due to the limited space. In terms of methodological development,
we focused on developing the first BN model for sparse count data, establishing identifiability theory, and designing
effective structural learning algorithm. Introducing latent factors that account for latent confounders would necessarily
complicate the model, theory, and computation, and may obscure the contributions of different components (zero-
inflation, sparsity prior, tempered MCMC algorithm) of our method to the good empirical performance. Focusing
on cases without confounders allowed us to compare with competing methods for count data that do not account for
confounders. That being said, it will be very interesting to extend our current work in this direction.
**[R3] Existing greedy algorithms.** We agree that Chickering's paper is indeed very relevant to this paper and should
have been mentioned in the paper. There are, however, several fundamental differences between the convergence of our
algorithm and that of Chickering's, which led us to the claim "unlike heuristic/greedy search algorithms, MCMC is
theoretically guaranteed to converge to its stationary distribution". (1) Our algorithm requires a sufficient number of
MCMC iterations whereas Chickering's requires a large sample size. In practice, a large enough sample size (relatively
to the super-exponential size of DAG space) is often infeasible and expensive to obtain, while it is comparatively
much easier and cheaper to increase the size of MCMC and its (lack of) practical convergence can be monitored
by various diagnostics (e.g. Gelman-Rubin's potential scale reduction factor). (2) Chickering's method requires
faithfulness whereas ours doesn't. Faithfulness can be violated with a limited sample size (see e.g., Ulher et al., 2013
*Ann. Stat.*) and/or in an equilibrium-maintaining system such as a biological system. Intuitively, while both Gaussian
and multinomial can have "accidental" cancellation of positive and negative effects (e.g., in a feedforward loop of
exercise, body temperature, and sweating) and hence become unfaithful, the proposed ZIPBN doesn't allow such
cancellation because of its count nature (also see e.g., Park and Park 2019, *AISTATS*). (3) We focus on identifying
individual BN whereas Chickering's focuses on identifying Markov equivalence class. (4) Greedy search algorithm
converges to a point estimate whereas MCMC-based method converges to the posterior distribution which allows for
finite-sample statistical inference (e.g., edge inclusion probability, FDR control) of the estimated BN.
**[R3] Non-sparse prior.** A common assumption for BN structure learning is sparsity, which is induced via sparse graph
priors in ZIPBN and sparse skeletons in ODS/MRS. For all methods, the assumed sparsity level can significantly affect
performance. ZIPBN and many other Bayesian methods do not use a uniform graph prior (a special case of Erdös-Rényi
with probability 0.5) because it favors dense graphs: (i) Erdös-Rényi is a well-known dense random graph and (ii) there
are far more dense graphs $|\boldsymbol{E}| = O(p^2)$ than sparse graphs $|\boldsymbol{E}| = O(p)$ (e.g., a graph with half the size of a complete
graph is still very dense). So as expected, when a dense uniform prior was adopted, FDR of ZIPBN went from 0.18 to
0.59 (n=500,p=50). However, same goes for ODS/MRS when skeleton learning cutoff is chosen to favor dense graphs.
**[All] Computation speed.** The worst-case per iteration cost is $O(np^2)$ (mainly the likelihood evaluation), which is
reduced to $O(\max(n,p)p)$ for sparse networks (i.e., $|\boldsymbol{E}| = O(p)$). The CPU time for real pathway analysis was 1.7hrs.
**[All] Clarifications.** $p_s$ was chosen to be 10% and the algorithm appears not sensitive in that choice. We gently point
out that Theorem 1 does hold for *all* networks because we proved that *any* two Markov equivalent networks with
different structures (i.e., $\boldsymbol{E}' \neq \boldsymbol{E}$, line 3 of the proof) have to have different distributions (note that two non-Markov-
equivalent BNs are trivially not distribution equivalent, e.g., Spirtes et al. 2000). FDR was controlled at 1% (line 286).
We will improve the description of priors, MCMC, intuition of non-identifiable BNs, notations, and broader impacts.

[Meta-Review · NeurIPS 2020]

All of the reviewers agree that this paper is both theoretically and modeling-wise a solid contribution to NeurIPS. My only concerns are that some of the author rebuttal points have not made it into the paper -- all of them should be added I think, in particular the related work (extended), the causal sufficiency clarification, and the run times.